# DataRater: Meta-Learned Dataset Curation

**Dan A. Calian**[*]  **Gregory Farquhar**[*]  **Iurii Kemaev**[*]  **Luisa M. Zintgraf**[*]
**Matteo Hessel**  **Jeremy Shar**  **Junhyuk Oh**  **András György**  **Tom Schaul**
**Jeffrey Dean**  **Hado van Hasselt**  **David Silver**

Google DeepMind

## Abstract

The quality of foundation models depends heavily on their training data. Consequently, great efforts have been put into dataset curation. Yet most approaches rely on manual tuning of coarse-grained mixtures of large buckets of data, or filtering by hand-crafted heuristics. An approach that is ultimately more scalable (let alone more satisfying) is to *learn* which data is actually valuable for training. This type of meta-learning could allow more sophisticated, fine-grained, and effective curation. Our proposed *DataRater* is an instance of this idea. It estimates the value of training on any particular data point. This is done by meta-learning using 'meta-gradients', with the objective of improving training efficiency on held out data. In extensive experiments across a range of model scales and datasets, we find that using our DataRater to filter data is highly effective, resulting in significantly improved compute efficiency.

## 1 Introduction

It is widely acknowledged in machine learning that the quality of a model fundamentally depends on the data used to train it. In the era of large foundation models, an enormous quantity of data, encompassing a wide spectrum of sources and reliability, is ingested for pre-training. Careful filtering and curation of this data has repeatedly proven critical for efficiency and capability [Hoffmann et al., 2022, Llama 3 Authors, 2024, Parmar et al., 2024]. Despite the vast scale of this challenge, laborious human curation approaches are the status quo: in general, data is filtered by hand-crafted heuristics, and final coarse-grained mixtures across sources are manually tuned.

In the near term, these data challenges will only become more acute: the next frontier is synthetic data, which can be generated in unbounded quantity, but can be biased, redundant, or otherwise of low quality. Hence it will be critical to employ systems able to ingest arbitrary amounts and types of unfiltered data, while processing this merged stream of data so as to maximize learning efficiency.

Addressing this necessitates automated methods for filtering and optimising the data stream. A highly scalable filtering method would let humans specify *what* they want at the end of training (e.g., in terms of a validation loss), rather than specifying *how* to achieve it (i.e., in terms of manually curating the data stream). This is precisely what *meta-learning*, as a solution method, offers – and is the approach we take in our work. Instead of manually specifying the filtering process, meta-learning enables us to automatically learn the criteria by which to filter or mix the data stream in a data-driven way. If we let it, the data can reveal its own value.

To this end, we introduce the DataRater, a method that estimates the value of any particular piece of data for training: the DataRater assigns a (meta-learned) preference weight to any data item, which can be used for filtering or re-weighting. It is grounded in a very simple meta objective: improving

---

[*]Equal contribution. Correspondence to {`dancalian`, `gregfar`, `iukemaev`, `zintgraf`}`@google.com`.

39th Conference on Neural Information Processing Systems (NeurIPS 2025).

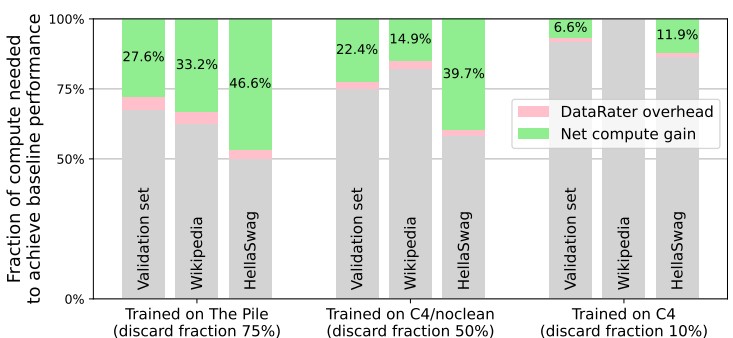

Figure 1: **Compute needed to achieve baseline performance using the DataRater.** A baseline 1B model is trained on the unfiltered dataset, while a second 1B model is trained analogously on the same dataset, but filtered by a DataRater. The x-axis states the underlying dataset, with 3 evaluation metrics each, while the y-axis shows the fraction of compute needed to match the baseline in grey. The overhead of using the DataRater for filtering online is shown in pink. To calculate this metric we convert both the LLM training step cost and the DR inference step costs into FLOPs (while also accounting for batch-size oversampling). 'Validation set' refers to the respective validation set of the underlying dataset. Net compute gain is shown in green. The figure shows that filtering data using the DataRater results in significant overall compute gains for lower quality datasets like the `Pile` & `C4/noclean`.

training efficiency on held out data (Section 2). The DataRater is trained using meta-gradients, which makes it highly sample-efficient compared to black box methods that see the consequence of the data, but do not see the function connecting data to performance.It is effective empirically at reducing training compute for matching performance, especially for filtering low quality training datasets (Section 3). Finally, the principles underlying the DataRater open a path toward further gains and use-cases, such as online adaptation during training, robustness to shifts in the data distribution, or tailoring data streams to narrowly targeted preferences, as discussed in Appendix A.

The main contributions of this work are:

- **DataRater framework.** We introduce the DataRater, a novel meta-learning framework designed to estimate the value of individual data points for training foundation models, relative to a specified meta-objective. Our approach aims to automate dataset curation by meta-learning the value of data.
- **Scalable meta objective.** The DataRater model is optimised using meta-gradients on a simple-to-specify objective of improving training efficiency on held-out data (Section 2).
- **Significant compute efficiency and performance gains.** Extensive experiments demonstrate that DataRater-filtered data substantially reduces training FLOPS (achieving up to $46.6\%$ net compute gain, Figure 1) and frequently improves final performance for language models across diverse pre-training corpora (e.g., the `Pile`, `C4/noclean`), as illustrated in Figure 6.
- **Cross-scale generalisation of data valuation.** Critically, a DataRater model meta-trained with fixed-scale inner models (400M parameters) effectively generalizes its learned data valuation policy to enhance training across a spectrum of target model scales (50M to 1B parameters, also Figure 6). Moreover, optimal data discard proportions are shown to be consistent across these model scales (Figure 4).
- **Insight into data quality.** Our analysis (Section 3) reveals that the DataRater learns to identify and down-weight data that aligns with human intuitions of low quality, such as incorrect text encodings, OCR errors, and irrelevant content.

## 2 Meta-Learned Dataset Curation

In this section we formally introduce our framework and derive our method.

**The filtering problem.** Suppose our aim is to develop a predictor over an input set $\mathcal{D}$, for example, $\mathcal{D}$ can be the set of all finite sequences of tokens $\mathbf{x} = (x_0, x_1, \ldots, x_{|\mathbf{x}|})$ over a token set. To do this, we are given a training set $\mathcal{D}_{\text{train}} \subset \mathcal{D}$ on which we train a foundation model with a loss function $\ell$, and we aim to use our model on a possibly different dataset $\mathcal{D}_{\text{test}} \subset \mathcal{D}$ and loss function $L$. For

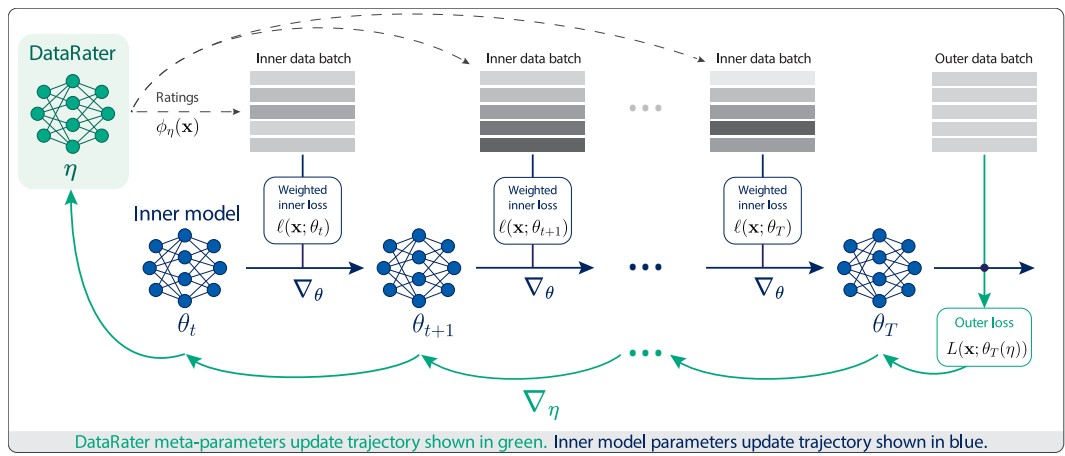

Figure 2: **DataRater meta-learning schematic.** This diagram shows how the DataRater is updated using meta-gradients to minimise an outer loss, by back-propagating through multiple inner model updates (we experimented with up to 8 inner updates) over weighted inner data batches. For more details on the implementation see Algorithm 1.

---

**Algorithm 1** Meta-learning a DataRater ($\phi_\eta$).

1: **Inputs:** Training dataset $\mathcal{D}_{\text{train}}$; testing dataset $\mathcal{D}_{\text{test}}$; DataRater $\phi_\eta$ with meta-parameters $\eta$ and meta-optimiser $H$; Inner model $f_\theta^i$ parameterised by $\theta^i$ with optimiser $\Theta$; $K$ outer update steps; $T$ inner model update steps; inner batch size $N_{\text{inner}}$, and outer batch size $N_{\text{outer}}$; inner loss function $\ell(\theta; \mathbf{x})$ and outer loss function $L(\theta; \mathbf{x})$; number of inner models $N_{\text{models}}$.

2: $\eta_0 \leftarrow \text{INITMETAPARAMS}()$                   ▷ Initialise DataRater.
3: **for** $i = 0 \dots N_{\text{models}} - 1$ **do**              ▷ Initialise inner models.
4:     $\theta_0^i \leftarrow \text{INITPARAMS}()$

5: **for** $k = 0 \dots K - 1$ **do**                   ▷ Outer loop ($\eta$).
6:     **for** $i = 0 \dots N_{\text{models}} - 1$ **do**
7:         $\theta_{k+1}^i = \text{UPDATEINNERMODEL}(\theta_k^i, \eta_k)$      ▷ Update inner model for $T$ steps.
8:         $\mathcal{B}_k \sim \mathcal{P}_{N_{\text{outer}}}(\mathcal{D}_{\text{test}})$             ▷ Sample outer batch.
9:         $g_k^i = \frac{1}{N_{\text{outer}}} \sum_{\mathbf{x} \in \mathcal{B}_k} \nabla_\eta L(\mathbf{x}; \theta_{k+1}^i(\eta_k))$     ▷ Form meta-gradient.
10:         $\bar{\eta}^i = H(\eta_k, g_k^i)$       ▷ Compute per-model meta-parameter update.
11:     $\eta_{k+1} = \frac{1}{N_{\text{models}}} \sum_i \bar{\eta}^i$             ▷ Update DataRater.
12: **Return** $\eta_K$              ▷ Optimised DataRater meta-parameters.

13: **function** $\text{UPDATEINNERMODEL}(\theta_0, \eta)$
14:     **for** $t = 0 \dots T - 1$ **do**                 ▷ Inner loop ($\theta$).
15:         $\mathcal{B}_t \sim \mathcal{P}_{N_{\text{inner}}}(\mathcal{D}_{\text{train}})$           ▷ Sample inner batch.
16:         $g_t = \sum_{\mathbf{x} \in \mathcal{B}_t} \nabla_\theta \ell(\mathbf{x}; \theta_t) \cdot \sigma_{\mathcal{B}_t}(\phi_\eta(\mathbf{x}))$    ▷ Form weighted gradient using data ratings.
17:         $\theta_{t+1} = \Theta(\theta_t, g_t)$              ▷ Update parameters.
18:     **Return** $\theta_T$           ▷ $T$-steps updated inner model parameters.

---

example, $\mathcal{D}_{\text{test}}$ can be a clean dataset for a problem while $\mathcal{D}_{\text{train}}$ can be its noisy version (e.g., see Figure 3 for a motivating toy example). Given a parameter vector $\theta$ of the predictor, the loss of a sample $\mathbf{x} \in \mathcal{D}$ is $\ell(\mathbf{x}; \theta)$ for the training and $L(\mathbf{x}; \theta)$ for the test problem.

In this setting a typical gradient-based learning algorithm, $\mathcal{A}$, performs $T$ steps on random minibatches $\mathcal{B}_t = (\mathbf{x}_t^1, \dots, \mathbf{x}_t^N)$ selected from the training data $\mathcal{D}_{\text{train}}$ for each step $t = 0, \dots, T - 1$:

$$\theta_{t+1}, s_{t+1} = \Theta(\theta_t, s_t, g_t) \qquad \text{with} \qquad g_t = \frac{1}{N} \sum_{i=1}^{N} \nabla_\theta \ell(\mathbf{x}_i; \theta_t); \qquad (1)$$

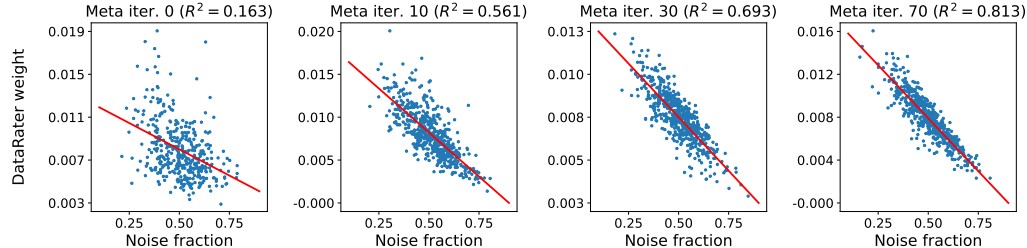

Figure 3: **Motivating toy example: DataRater learns to weight examples in proportion to their corruption level.** Given two datasets, $\mathcal{D}$ and $\hat{\mathcal{D}}$, drawn from the same distribution, but where sequences in $\hat{\mathcal{D}}$ have been corrupted with token-level noise of various levels, from 0 (clean data) to 1 (fully random tokens). The DataRater weights correlate increasingly well with the noise fraction, showing that the DataRater recognises that corrupt sequences are worse for training than clean ones.

where $\Theta$ is the update function of the learning algorithm $\mathcal{A}$, and $s_t$ denotes its state at step $t$. After $T$ steps, this process leads to a (random) parameter vector $\theta_T$; we will denote this by $\theta_T(\mathcal{D}_{\text{train}}) \sim \mathcal{A}_T(\mathcal{D}_{\text{train}})$, where in the notation we omit the dependence on the initial parameter vector $\theta_0$ and optimiser state $s_0$ for simplicity. The performance of (one run of) the algorithm is measured on the test data by

$$L(\theta_T(\mathcal{D}_{\text{train}}); \mathcal{D}_{\text{test}}) = \mathbb{E}_{\mathbf{x} \sim \mathcal{D}_{\text{test}}} \left[ L(\mathbf{x}; \theta_T(\mathcal{D}_{\text{train}})) \right], \tag{2}$$

where $\mathbf{x} \sim \mathcal{D}_{\text{test}}$ denotes a uniform distribution over the test dataset $\mathcal{D}_{\text{test}}$ (and could be replaced by an arbitrary test distribution over $\mathcal{D}$).

The goal of a *filtering process* is to select a subset of training data that leads to the smallest test error,

$$\mathcal{D}_{\text{train}}^* = \operatorname*{argmin}_{\bar{\mathcal{D}} \subseteq \mathcal{D}_{\text{train}}} \mathbb{E} \left[ L(\theta_T(\bar{\mathcal{D}}); \mathcal{D}_{\text{test}}) \right], \tag{3}$$

where the expectation is taken over the randomness of the learning algorithm $\mathcal{A}$.

Note that even when the training and test distributions are the same, $\mathcal{D}_{\text{train}} = \mathcal{D}_{\text{test}}$, it might be more efficient to train on a subset of $\mathcal{D}_{\text{train}}$ only, as the learning algorithm typically does not find the optimal parameter $\theta^*$ and/or may train faster on a well-selected subset of the data (e.g., omitting data points which are already learnt).

**The DataRater algorithm.** Discrete subsampling of which data to keep is known to be a non-deterministic polynomial-time hard problem, even for linear models [Davis et al., 1997, Natarajan, 1995]. Like many others [e.g. Liu et al., 2018], we solve a continuous relaxation instead.

We proceed in three steps. First, instead of optimising over the best subset of $\mathcal{D}_{\text{train}}$ in (3), we optimise over which data points (from a larger minibatch) to include in the gradient calculation $g_t$ at step $t$; this is equivalent to introducing a binary weight for each data point in the minibatch.

Secondly, we perform a continuous relaxation, replacing the binary weights with continuous weights in $[0, 1]$. Thus, instead of (1), we calculate a weighted gradient for the minibatch in the form $g_t = \sum_{i=1}^{N} w(\mathbf{x}_i) \nabla_\theta \ell(\mathbf{x}_i; \theta_t)$, where $w(\mathbf{x}) \in [0, 1]$ are the continuous weights. To keep the gradient of similar order, we enforce that the weights sum up to 1 in each batch.

Finally, to represent the weights we use function approximation, rather than tabulating a weight per data point: we introduce a DataRater model (i.e., a score function) $\phi_\eta : \mathcal{D} \to \mathbb{R}$, parameterised by $\eta$, which measures the value of each data point; we obtain the normalised weights for every point in a batch through a softmax preference function: $\sigma_{\mathcal{B}_t}(\phi_\eta(\mathbf{x})) = \frac{e^{\phi_\eta(\mathbf{x})}}{\sum_{\mathbf{x}' \in \mathcal{B}_t} e^{\phi_\eta(\mathbf{x}')}}$.

Thus, the filtering weights specified by $\eta$ induce a training algorithm $\mathcal{A}(\mathcal{D}_{\text{train}}; \eta)$ that for $t = 0, \ldots, T-1$ does: sample $\mathcal{B}_t = (\mathbf{x}_1, \ldots, \mathbf{x}_N)$ uniformly at random from $\mathcal{D}_{\text{train}}$ and compute

$$\theta_{t+1}, s_{t+1} = \Theta(\theta_t, s_t, g_t) \qquad \text{with} \qquad g_t = \sum_{i=1}^{N} \nabla_\theta \ell(\mathbf{x}_i; \theta_t) \cdot \sigma_{\mathcal{B}_t}(\phi_\eta(\mathbf{x}_i)). \tag{4}$$

We denote the resulting (random) parameter vector by $\theta_T(\eta) \sim \mathcal{A}(\mathcal{D}_{\text{train}}; \eta)$.

To optimise the filtering algorithm, we want to find the DataRater parameters $\eta$ minimizing the expected test error:

$$\eta^* = \underset{\eta}{\arg\min} \, \mathbb{E}[L(\theta_T(\eta); \mathcal{D}_{\text{test}})]$$

where again the expectation is taken over the randomness of the algorithm.

**Meta-optimisation.** To optimise the parameters, $\eta$, of a DataRater model approximately, we use a *meta-gradient method* [Xu et al., 2018]: $\eta$ is optimised via a stochastic gradient method, where the gradient of the loss $L$ (which is called *outer loss* in this context) with respect to $\eta$ is computed by back-propagating through multiple optimiser updates for $\theta$ with respect to the so-called *inner loss* $\ell$, as illustrated in Figure 2; the method is given formally in Algorithm 1[2].

**Meta-learning objective.** We described the general DataRater framework above. For the remainder of the paper, including all experimental evaluation, we choose to focus on the most widely applicable objective: *where the train and test datasets share the same distribution*. This choice does not enforce that one must be interested in optimising for some specific downstream task, but rather, it assumes that one simply wants to maximise training efficiency with respect to a given dataset.

So, the held out data, on which the DataRater outer loss is defined, is a disjoint subset of the input training data. Put differently, our objective is to produce a curated variant $\mathcal{D}^*$ of a given original dataset, so that training on it results in *faster learning on the original training dataset*. In this case, the inner ($\ell$) and outer ($L$) loss share the same functional form, i.e., cross entropy loss w.r.t the next token prediction, $\ell(\mathbf{x}; \theta) = L(\mathbf{x}; \theta) = -\log \mathrm{P}_\theta(\mathbf{x}) = -\sum_{i=0}^{|\mathbf{x}|} \log \mathrm{P}_\theta(x_i | x_{i-1}, \ldots, x_0)$.

**Implementation.** We instantiate the DataRater as a non-causal transformer and optimise its parameters $\eta$ using meta-gradient descent; i.e. back-propagating through the unrolled optimisation of the inner model. We back-propagate through a truncated window of 2 inner model updates to compute the meta-gradient[3]. This process, handled implicitly by automatic differentiation, involves operations with second-order derivatives, such as Hessian matrices, which in practice are computationally expensive. To make this computation feasible, we use the scalable bilevel optimisation reparameterisation of Kemaev et al. [2025] called MixFlow-MG. This method exploits implicit symmetries present in bilevel gradient optimisation problems and uses mixed-mode differentiation to significantly reduce RAM (HBM) usage. In particular, we use all the core techniques from MixFlow-MG, including block-level rematerialisation with saving inner-level gradients, and mixed-mode differentiation itself. This enables us to optimise DataRater meta-parameters efficiently – even for 50M DataRater models and 400M inner models, while back-propagating through multiple inner model update steps.

To stabilise meta-gradient computation we: (1) use a population of inner models; (2) pass the meta-gradient for each inner model through an individual Adam optimiser [Kingma and Ba, 2015], averaging the resulting meta-gradient updates; (3) periodically reinitialise inner models to stratify learning progress. In the appendix in Figure 12 we plot the norm of the meta-gradient updates over the course of meta-training, for each inner model, showing that they remain stable and bounded. Furthermore, in Figure 13 we show that the temporal auto-correlation of DataRater scores approaches $> 0.95$ after a few thousand meta-update steps, indicating that meta-training is converging.

**Data curation.** For data curation, with a trained DataRater, we use top-$K$ filtering at the batch-level: instead of weighting data, we remove data with low predicted value. To implement top-K filtering at the batch level, for a given target discard fraction $\rho \in [0, 1)$, and a nominal batch size $N$, we upsample a larger batch of data points sized $\frac{N}{1-\rho}$, score them using a DataRater, and filter out the bottom $(100 \cdot \rho)$-percent of samples.

DataRater scores can also be equivalently used to select data points individually – without requiring access to a batch of data. This requires access to the CDF of the ratings distribution, $F_\Phi : \mathbb{R} \to [0, 1]$. To emulate the batch-level decision for top-$K$ filtering given a batch size of $B$, for a data point $\mathbf{x}$ with $p = F_\Phi(\phi(\mathbf{x}))$, we can calculate the probability of keeping it as $\mathrm{P}_{\text{accept}}(\mathbf{x}) = \sum_{s=0}^{K-1} \binom{B-1}{s}(1 - p)^s p^{B-s-1}$ (since we keep a data point if select at most $K - 1$ "better" points in a batch). This is

---

[2]Pseudo-code omits optimiser states and periodic resetting of inner model parameters.

[3]We experimented with 1, 2, 4 and 8 inner model updates, and found that using 2 inner updates results in very similar DataRater performance as using more.

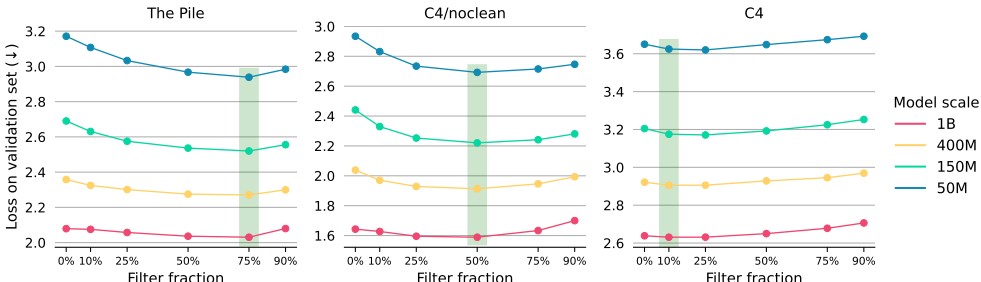

Figure 4: **Loss on validation subsets, over multiple model scales, as a function of the percentage of data filtered out according to DataRater models (one DataRater model is trained for each dataset).** For each underlying dataset (the `Pile`, `C4/noclean` and `C4`), the data filtering proportion resulting in the best validation set performance remains constant across evaluated model scales (highlighted in green).

useful for building massively parallel data filtering pipelines using, e.g., Apache Beam [Dean and Ghemawat, 2004], as filtering decisions can be made locally.

## 3 Experimental Results

**Evaluation protocol.** To evaluate the effectiveness of our method in curating a dataset we perform the following three main steps: (1) meta-learn a DataRater for the input dataset $\mathcal{D}$; (2) use the trained DataRater to curate it, forming $\mathcal{D}^*$; (3) train language models (of 50M, 150M, 400M and 1B sizes) from random initialisation on the curated dataset and subsequently evaluate their performance.

**Datasets.** Our experiments utilise three datasets selected for their varying degrees of pre-filtering: `C4` [Raffel et al., 2020], the most stringently filtered; `C4/noclean`, a less-filtered version of `C4`; and the `Pile` [Gao et al., 2020], representing the least filtered data. We we use the English subsets of C4, i.e. `C4` and `C4/noclean` refer to `c4/en` and `c4/en.noclean` from `huggingface.co` respectively.

**Models and training.** All inner language models and DataRater models are based on the Chinchilla [Hoffmann et al., 2022] transformer architectures [Vaswani et al., 2017]. Models are trained from random initialisation, adhering to the Chinchilla training protocols and token budgets; i.e. the learning algorithm $\mathcal{A}$ (and optimiser update function $\Theta$) is defined by the Chinchilla scaling laws.

**DataRater configuration.** For each of the three datasets, we train a distinct 50M parameter non-causal transformer as the DataRater model. Each DataRater is meta-learned using a population of eight 400M parameter inner language models (see Appendix A for meta-learning checkpoint selection details). Importantly, for each input dataset, the inner models and their corresponding DataRater model are optimised on disjoint subsets of that dataset's training data.

**Evaluation.** We measure negative log likelihood (NLL) on the validation set of each of the three input datasets, as well as on (English-language) Wikipedia. We measure accuracy on the following downstream tasks: HellaSwag [Zellers et al., 2019], SIQA [Sap et al., 2019], PIQA [Bisk et al., 2020], ARC Easy [Clark et al., 2018] and Commonsense QA [Talmor et al., 2019], which we selected as they provide useful signal at the model scales we use (i.e. benchmark performance increases during the course of training and across model scales). In Figure 1 we plot the fraction of compute needed to match baseline performance for 1B models. To calculate this metric we convert both the LLM training step cost into FLOPs as well as the DR inference step cost into FLOPs, while also accounting for batch-size oversampling prior to DR online filtering. We leave this metric undefined for cases where training on the filtered datasets does not reach baseline performance (e.g. for some cases on the `C4` dataset).

**Can the DataRater accelerate learning?** The DataRater indeed accelerates learning, resulting in considerable FLOPS savings for lower quality datasets like the `Pile` and `C4/noclean`. For 1B models, Figure 1 shows precisely the amount of compute saved by using the DataRater, when compared to baseline training, while also showing the compute overhead due to DataRater filtering. Figure 5 shows performance metrics during training of 1B models. This shows that, particularly for

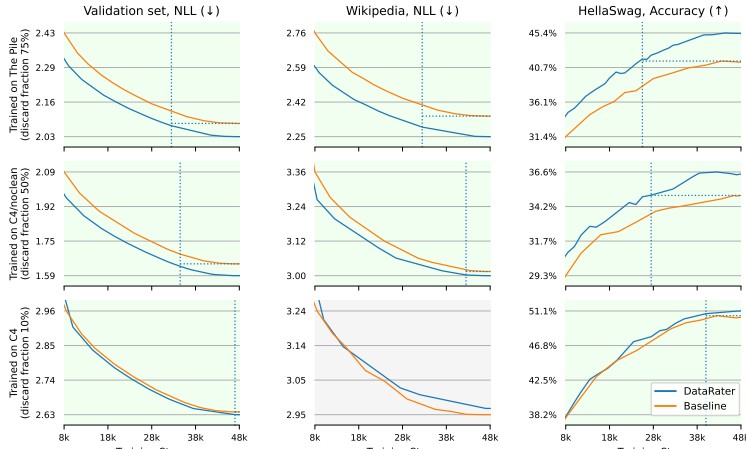

Figure 5: **Learning curves of 1B models trained on three separate underlying datasets, with DataRater filtering and without**. On both `C4/noclean` and the `Pile`, models trained on DataRater filtered data match the baseline's final performance while using considerably fewer training steps, while also resulting in improved final performance. Using a DataRater to filter `C4`, which is a considerably higher quality dataset, results in minor performance improvements on average (i.e. see Table 2 in appendix for full results) while exhibiting trade-offs in downstream evaluations as exemplified here.

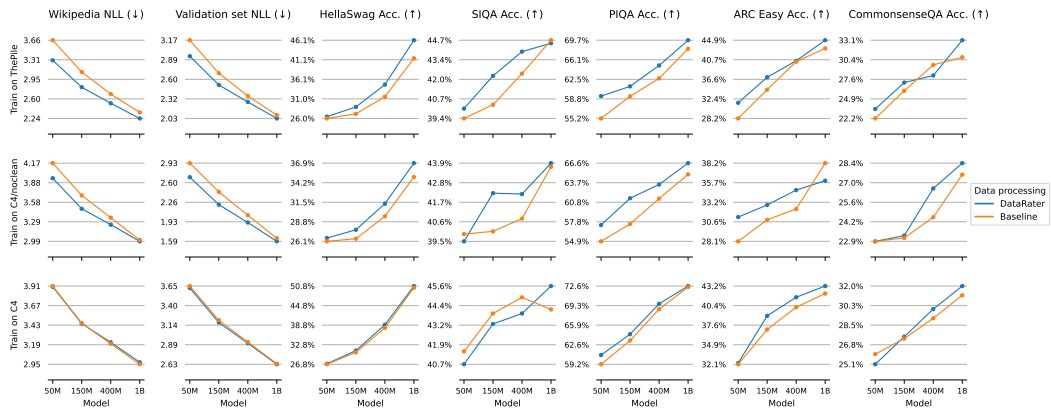

Figure 6: **DataRater filtering results in performance improvements across the majority of model scales and downstream tasks.** Each individual plot shows downstream evaluation performance (measured either as accuracy or NLL) as a function of model scale. Each row corresponds to one of three separate training datasets, while each column corresponds to a specific downstream evaluation.

lower-quality datasets, using DataRater filtered datasets often results in improved final performance, and not just in faster learning.

Of course, training the DataRater model itself must also be accounted for. Meta-training a DataRater requires approximately $58.4\%$ of the FLOPS needed to train a single 1B LLM. Given that the filtered dataset produced by the DataRater will be used to train numerous other LLMs, of possibly much larger scale, we argue that the cost of training the DataRater can be amortised away.

In Section A in the appendix we compare the DataRater approach with a perplexity-based filtering method adapted from prior work [Ankner et al., 2025], showing that the DataRater results in better performance in a large majority of evaluations, i.e. in 16 out of 21 cases.

**How much data should be removed?** To select the best discard proportion for each dataset, we run a sweep over 5 candidate discard proportions ($10\%$, $25\%$, $50\%$, $75\%$ and $90\%$), using the smallest model (50M) and select the discard proportion resulting in the best *validation set* NLL. In Figure 4

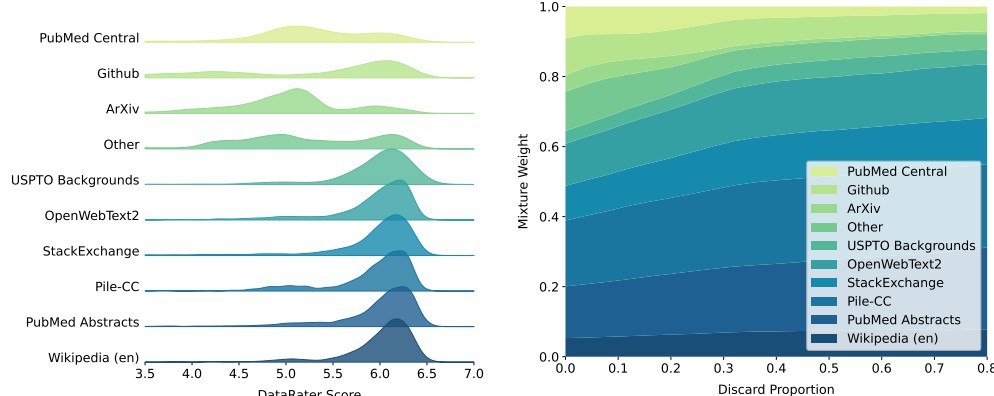

Figure 7: (a) Distribution of DataRater scores on data subsets from the `Pile`. (b) DataRater induced mixture weights for the `Pile` data subsets as a function of discard proportion (from 0% to 80%).

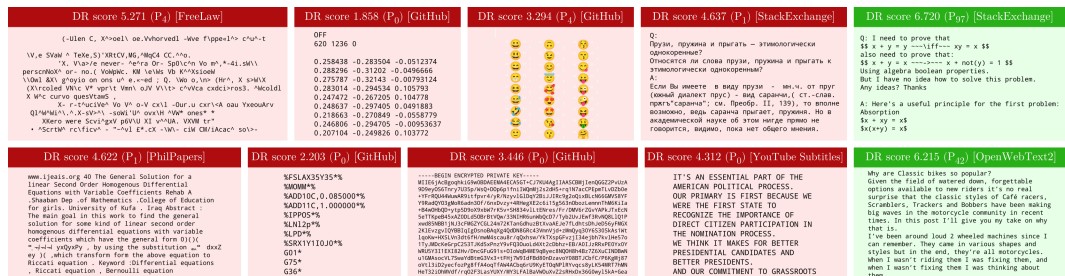

Figure 8: Qualitative samples from the `Pile` annotated with their DataRater score, score percentile and provenance.

we show that using the smallest model size is sufficient, as the best discard proportion is shared across model sizes. The optimal value depends on the underlying data quality: for `C4` we drop $10\%$, for `C4/noclean` $50\%$ and for the `Pile` $75\%$.

**Is the DataRater robust to model scale?** We trained a single DataRater per dataset, with a fixed inner model size of 400M. The DataRater generalises across model scales: in experiments over 3 input datasets over 4 model scales and 7 metrics, $73/84$ showed improvements (see Figure 6). For the lower quality datasets, `C4/noclean` and the `Pile`, the performance improvements are consistent over model scale for NLL metrics and HellaSwag, whereas downstream evaluations have more variance.

**What does the DataRater learn to do?** The DataRater learns to assign fine-grained ratings to different data subsets, as shown in Figure 7 (a). This plot shows the distribution of raw DataRater scores (i.e., $\phi_\eta(\mathbf{x})$) for multiple subsets of the the `Pile`; most subsets are assigned a heavy left tail – identifying data points which should be discarded even at very low discard proportions. In effect, the DataRater also learns to re-weight data at the mixture level, as shown in Figure 7 (b) for increasing discard proportions.

The DataRater also learns to identify low quality data; i.e. samples which are assigned low ratings by the DataRater on the `Pile` dataset tend to be low quality. As exemplified in Figure 8, these include incorrect text encodings, OCR errors, large amounts of whitespace, non-standard non-printable characters (e.g., in the FreeLaw subset of the `Pile`), high entropy text such as lists and tables of numerical or string data as well as private SSH keys, English language in all capital letters, multilingual data (the `Pile` contains over 97% English data [Gao et al., 2020]), etc.

In Section A in the appendix we perform a correlation analysis between the DataRater score and common language filtering heuristics, with results shown in Figure 14. The DataRater score is most positively correlated with the number of packed sub-sequences as well as with various metrics of text length; it is most negatively correlated with the proportion of non-alphanumeric characters, of punctuation and of unique characters. An OLS model results in an $R^2$ of 0.766. As many of the heuristics are collinear, we fit a cross-validated Lasso ($L1$) model on Z-scored features to identify

a small but predictive subset of heuristics. The Lasso model explains 75.3% of the variance in the DataRater scores while using only 11 non-zero coefficients. We find that the DataRater score can mostly be explained by the number of packed sub-sequences, the proportion of non-alphanumeric characters, the word count, the sequence length, the packing density as well as a few other heuristics. Since neither regression model can perfectly predict the DataRater's score, it is likely the DataRater is identifying valuable patterns in the data that are not captured by these simple heuristics.

Further analysis about what the DataRater learns is presented in Appendix B.

## 4   Related Work

The curation of training data remains essential in ML and LLMs [Touvron et al., 2023, Zhou et al., 2023, Albalak et al., 2024]. Automated data selection strategies include heuristic filtering, learning-based strategies [Albalak et al., 2024, Jiachen Wang, 2024], and an emerging trend in bilevel optimisation methods [Pan et al., 2024, Wang et al., 2020, Shen et al., 2025, Dagréou et al., 2022].

**Heuristic Data Filtering.** Predefined heuristics to clean massive raw corpora remain prevalent. The C4 dataset [Raffel et al., 2020], for example, applied rules like language identification using simple classifiers (e.g., using fastText [Joulin et al., 2016]), removal of lines without terminal punctuation, and n-gram-based deduplication. More recent large-scale efforts such as FineWeb [Penedo et al., 2024] and Dolma [Soldaini et al., 2024] detail extensive, multi-stage pipelines encompassing URL filtering, quality heuristics (e.g., adapted from Gopher [Rae et al., 2021] or C4), content filtering for boilerplate text, navigation menus, error messages, personally identifiable information, toxic language, and various forms of deduplication using techniques like hashing (e.g., SHA1, MinHash with LSH), suffix arrays for exact substring matching, or Bloom filters. General categories of these heuristics, including document length constraints, symbol-to-word ratios, and boilerplate removal, are well-established [Albalak et al., 2024, Jiachen Wang, 2024]. While effective, these methods require manual tuning and may not consider less human-interpretable or intuitive features.

**Learned Data Selection Methods.** To move beyond fixed heuristics, various machine learning techniques have been proposed. Data valuation and influence-based techniques attempt to quantify the contribution of data points to model performance or capabilities. Foundational ideas include influence functions [Hampel, 1974, Koh and Liang, 2017], Cook's distance [Cook, 1977], and the Shapley value [Shapley et al., 1953, Ghorbani and Zou, 2019, Wang et al., 2025]. More recently, JST [Zhang et al., 2025] uses a two-stage valuation to refine the selection of high-quality data by initially leveraging identified low-quality data. PreSelect [Shum et al., 2025] quantifies "predictive strength" – how well data predicts downstream abilities – to guide selection. Relatedly, Gu et al. [2025] formulates data selection as a generalized optimal control problem, and solves it for LLMs approximately. Some approaches train classifiers to distinguish high-quality from low-quality text [Shum et al., 2025, Penedo et al., 2024]. Others leverage signals (such as perplexity scores) from pre-trained models [Wenzek et al., 2020], or use comparisons to baseline predictors [Lin et al., 2024, Mindermann et al., 2022, Sow et al., 2025]. Dynamic methods aim to select data during the training process itself; for example, GREATS [Wang et al., 2024] proposes an online batch selection algorithm using Taylor approximations of data quality. Our method shares the goal of learning the data value but uses meta-learning to directly optimise for the outcome of the learning process.

**Bilevel Optimisation and Meta-Learning for Data Selection.** The most analogous approaches to the DataRater frame data selection as a bilevel optimisation problem to learn a selection or weighting policy. Maclaurin et al. [2015] demonstrate computing exact gradients of validation performance by reversing learning dynamics, while many subsequent works (e.g., [Pedregosa, 2016, Lorraine et al., 2020]) use the implicit function theorem for efficient approximation. Differentiable Data Selection [Wang et al., 2020] uses a bilevel setup where a scorer network learns to weight data instances to optimise a model's performance, using gradient alignment as a reward. A similar bilevel weighting approach was used by [Guo et al., 2020] for safe semi-supervised learning. Others use similar bilevel frameworks to learn training data distributions under domain shift [Grangier et al., 2024] or to adaptively re-weight per-task gradients to improve multilingual training [Li and Gong, 2021].

Contemporaneously, SEAL [Shen et al., 2025] learns a 'data ranker' via bilevel optimisation to select fine-tuning data to enhance LLM safety while preserving utility; our formulation mathematically allows for arbitrary inner and outer losses, but, in this work, we focus on improving training efficiency

specifically. SEAL optimises the bilevel problem using a penalty method [Clarke, 1990], while we use exact meta-gradients. Concurrent work [Engstrom et al., 2025] similarly applies meta-gradients to optimise training configurations, including to select multi-modal data for CLIP [Radford et al., 2021] and instruction tuning data for Gemma-2B [Gemma Team, 2024] with a similar meta-objective as our method. Their method tracks one meta-parameter per data point, while we use function approximation to learn to assign value to arbitrary individual data points. We use all techniques from MixFlow-MG [Kemaev et al., 2025] (gradient checkpointing, mixed-differentiation, etc.) to efficiently compute meta-gradients, while their method introduces the 'REPLAY' algorithm for the same purpose.

## 5   Conclusion

This paper introduces the DataRater, a novel meta-learning approach for dataset curation that estimates the value of data to enhance the compute efficiency of training foundation models. By employing meta-gradients, our method is trained with the objective of improving training efficiency on held out data. Our experiments across various model scales and datasets demonstrate that our method is highly effective in filtering data, leading to significant compute efficiency improvements.

Our method can capture fine-grained distinctions in data quality, identifying and discarding less valuable data points which correlate with human intuitions of low quality text data. We also demonstrated robustness in compute efficiency improvements across different model sizes. The DataRater is particularly effective when the underlying dataset is low quality, so may be well suited to contexts where the quality of data is highly variable and human intuition struggles to easily define heuristics to estimate that value. Future work applying the DataRater to synthetic data or sensor data may therefore be particularly promising.

This approach offers a meaningful step towards automating and improving current dataset curation practices, which currently rely heavily on manual tuning and handcrafted heuristics. Our method provides a scalable and principled way to determine the value of individual data relative to the specified meta-objective, and shows considerable compute efficiency gains for filtering multiple real-world datasets when training on models of significant scale – thus providing a first proof of concept that meta-learning the data curation pipeline for modern LLMs is possible.

## Acknowledgements

The authors would like to express their gratitude to John E. Reid, Paul Michel, Katie Millican, Andrew Dai, Andrei A. Rusu, Jared Lichtarge, Chih-Kuan Yeh, Oriol Vinyals, the Gemini Data Team and the RL team for fruitful discussions, valuable feedback, and support throughout the project. The authors are also grateful to the anonymous reviewers for their insightful comments and constructive suggestions.

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

# A   Appendix

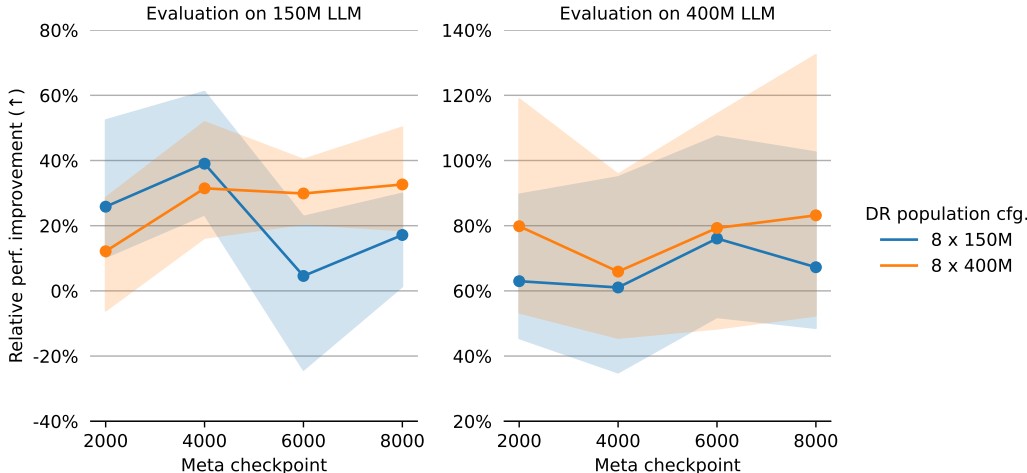

Figure 9: **Relative performance improvement induced by DataRater models trained on populations consisting of different sizes of inner models, as a function of the meta-training checkpoint.**
The underlying training dataset is `C4/noclean` and all evaluations are done with a $50\%$ discard fraction. The relative performance improvement is calculated with respect to a 1B baseline trained on the unfiltered `C4/noclean` dataset. The plots show the average (and $95\%$ CI) of the aggregate performance metric on HellaSwag, CommonsenseQA, PIQA, SIQA and Wikipedia. The DataRater trained with eight 400M inner models obtains the highest and most consistent performance across the two evaluated model scales and four meta-training checkpoints.

**Selecting the inner model scale and checkpoint.** As explained in the main paper, to meta-train a DataRater model, we approximately solve a bilevel optimization problem using stochastic gradient descent. We optimise a DataRater model's meta-parameters while computing meta-gradients using a population of individually trained inner language models.

We experimented with various hyper-parameter choices (including varying the meta-unroll length, the size of the population, etc.) and found that meta-evaluation performance depends mostly on the scale of the inner models in the population. In Figure 9 we show the relative performance improvement due to DataRater models trained on populations of inner models with different scales (150M and 400M). We perform this experiment only on the `C4/noclean` underlying dataset and use a fixed discard fraction of $50\%$.

In this plot (Fig. 9) we show relative performance improvement as a function of the meta-training checkpoint index (i.e., number of meta-update steps) and inner model scale. We calculate the relative performance improvement as the average percentage improvement in the raw metric (NLL for the `C4/noclean` validation set; accuracy for all the downstream tasks stated in the figure caption) of an LLM trained on the DataRater filtered data vs. a 1B baseline LLM (trained like-for-like on the unfiltered `C4/noclean` dataset).

We observe that the DataRater trained on a population of eight 400M inner models results in the best performance across the board. So, we choose this configuration (8 x 400M) and the meta-training checkpoint of 8000 steps to produce all results in the main paper body – including for experiments which use other underlying training datasets (`C4` and the `Pile`).

**Perplexity-based filtering.** We compare the DataRater to perplexity-based filtering, as perplexity is a common heuristic for assessing data quality that does not require choosing a specific validation set. For a fair and balanced comparison, all experiments in this section are conducted at the 150M model scale. We use a 50M DataRater model that was meta-trained on inner models of 150M scale (as opposed to the 400M inner models used for the results presented in the main paper).

We implement a perplexity-based filtering method similar to the one by Ankner et al. [2025]. To mirror the DataRater's online, batch-level filtering mechanism, we first train three separate 150M

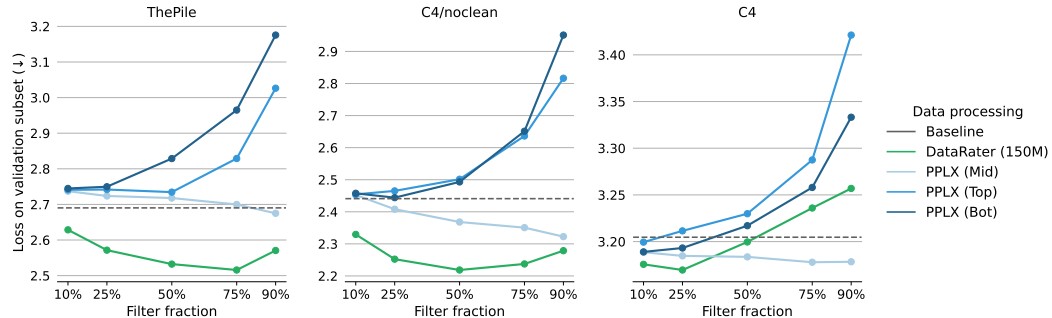

Figure 10: **Comparison of data filtering methods by sweeping over the filtering fraction**. The plots show validation loss as a function of the proportion of data filtered out for the DataRater (150M), there perplexity-based filtering baselines (PPLX-(Top, Mid, Bot)), and a no-filtering baselines. The PPLX (Mid) variant is the most effective of the perplexity-based methods, but the DataRater (150M) achieves the best performance across the majority of filtering fractions.

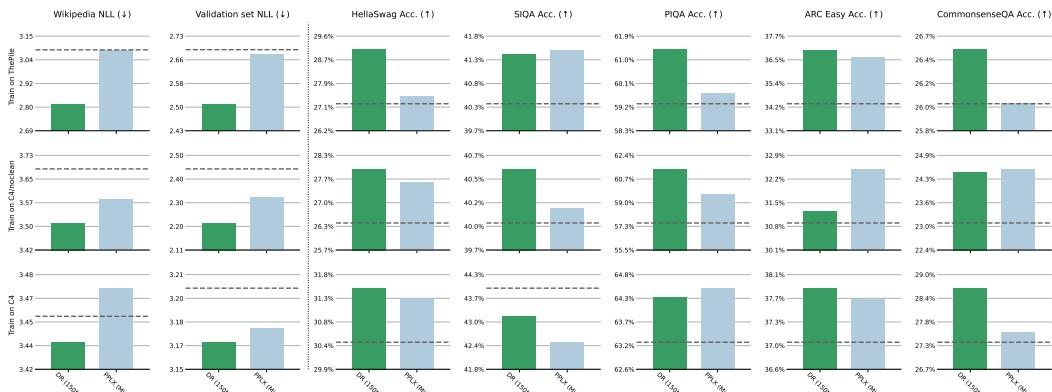

Figure 11: **Detailed comparison of DataRater (150M) to the best perplexity-based filter** PPLX (Mid). Baseline (no-filtering) performance is shown using dotted horizontal grey lines. This comparison uses the best filtering fractions for each method as identified from the sweep in Figure 10. Performance is evaluated across seven metrics for each of the three datasets. The DataRater (150M) results in the best performance on 16/21 evaluations, while PPLX (Mid) shows better performance on four.

baseline models on held-out subsets of each of the three training sets used throughout this paper. Each of these baseline models then acts as a scorer, assigning a perplexity score to each data point (i.e. packed tokenized text sequence). Filtering is then performed within each oversampled batch by selecting data based on these scores. We use the same oversampling mechanism for filtering as for the DataRater method. We test the three variants of perplexity-based filtering introduced in [Ankner et al., 2025]: keeping only the data with the lowest perplexity (PPLX (Bot)), the highest perplexity (PPLX (Top)), and the data from the middle of the perplexity distribution within each batch (PPLX (Mid)).

Figure 10 presents a sweep over different filtering fractions for the perplexity-based methods alongside the DataRater (150M) and the no-filtering baseline on the three datasets. From this experiment we observe that PPLX (Mid) is the most effective of the perplexity-based filtering variants. The best validation set performance is achieved by filtering data using the DataRater (150M), across the majority of the filtering fractions evaluated. Interestingly, filtering by perplexity can result in improvements when filtering C4 with very high filtering proportions. Based on this experiment, we select the best variant of perplexity-based filter as the PPLX (Mid) for the next comparison – together with the best filtering fractions: 90% for the PPLX (Mid) method on all three datasets, and 75%, 50% and 25% for the DataRater (150M) for the Pile, C4/noclean and C4 datasets respectively. In Figure 11 we present an extensive comparison of the selected methods on the seven evaluation metrics used in the main paper. This comparison shows that filtering using the DataRater still results

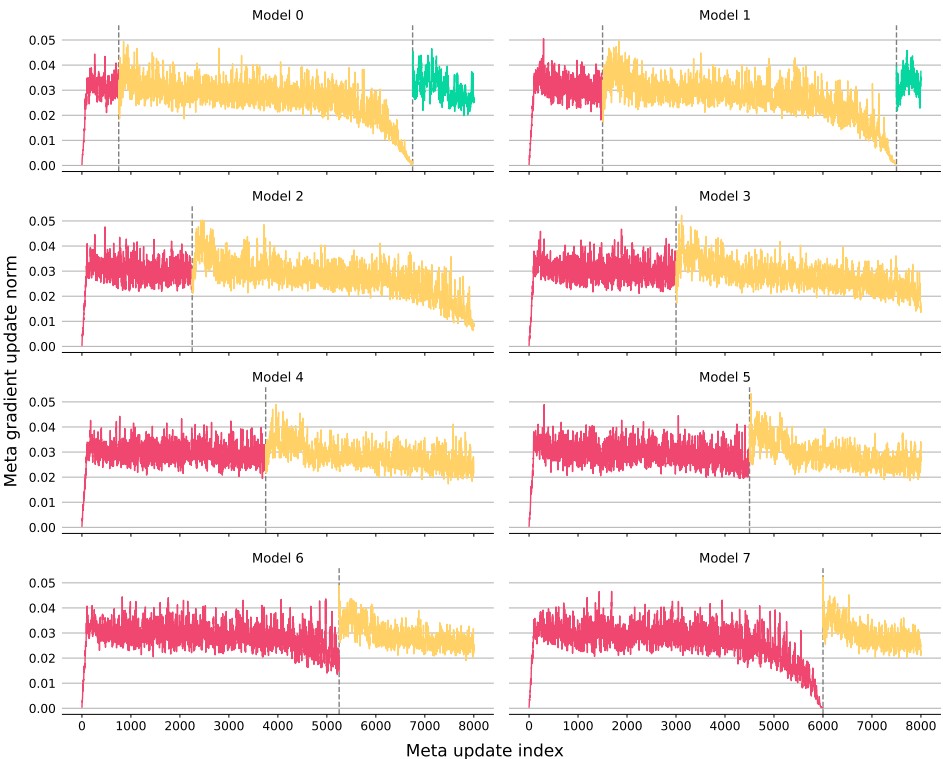

Figure 12: **Meta-gradient update norms through meta-training.** The plots show the L2 norm of meta updates originating from each individual inner model over the meta-training step counts of the `C4/noclean` 150M DataRater. The individual plots highlight when inner model parameters are reset by vertical dotted lines and by plotting individual lifetimes in different colours.

in the best performance across most evaluations (i.e. 16 out of 21 evaluations). Perplexity-based filtering (specifically, keeping the "middle" perplexity data from each oversampled batch) shows better performance on four evaluations (SIQA on the `Pile`, ARC Easy and CommonsenseQA on `C4/noclean`, and PIQA on `C4`), while best performance on SIQA results from not filtering `C4`.

**Meta-training stability.** As mentioned in the main paper body, we stabilise meta-gradient computation by using a population of inner models, passing their meta-gradients through individual Adam optimisers (averaging the resulting meta-gradient updates) and periodically reinitialising inner models. Since our goal is to use a trained and frozen DataRater to filter datasets for future downstream runs we require the DataRater to generalise across all stages of training – rather than to overfit to a particular inner model or to a particular stage of training. Having a population of inner models, whose parameters are reset occasionally, helps maintain diversity over the stage of training and promotes this type of generalisation. Our stabilisation strategies result in stable and bounded meta-updates as shown in Figure 12; the figure also highlights how inner models' lifetimes are stratified. Furthermore, Figure 13 shows the correlation between DataRater scores across meta-training, demonstrating that meta-training starts converging after a few thousand steps when scores start tending towards perfect correlation ($> 0.95$).

**Relation to heuristic filters.** We analyse the relationship between the DataRater score and common heuristics used for natural language data filtering. We adopted the heuristics used for filtering the `C4` dataset, sourced from the DataTrove library[4], along with other common metrics (e.g., non-alphanumeric character fraction, packed sub-sequence count, and word-level type-token ratio). A complete list of these heuristics is provided in Table 1.

To perform this analysis, we sampled $25600$ packed sequences from the `Pile`, calculated the DataRater score (using the the `Pile` 150M DR model) for each, and extracted all correspond-

---

[4]`https://github.com/huggingface/datatrove/blob/main/src/datatrove/pipeline/filters/c4_filters.py`

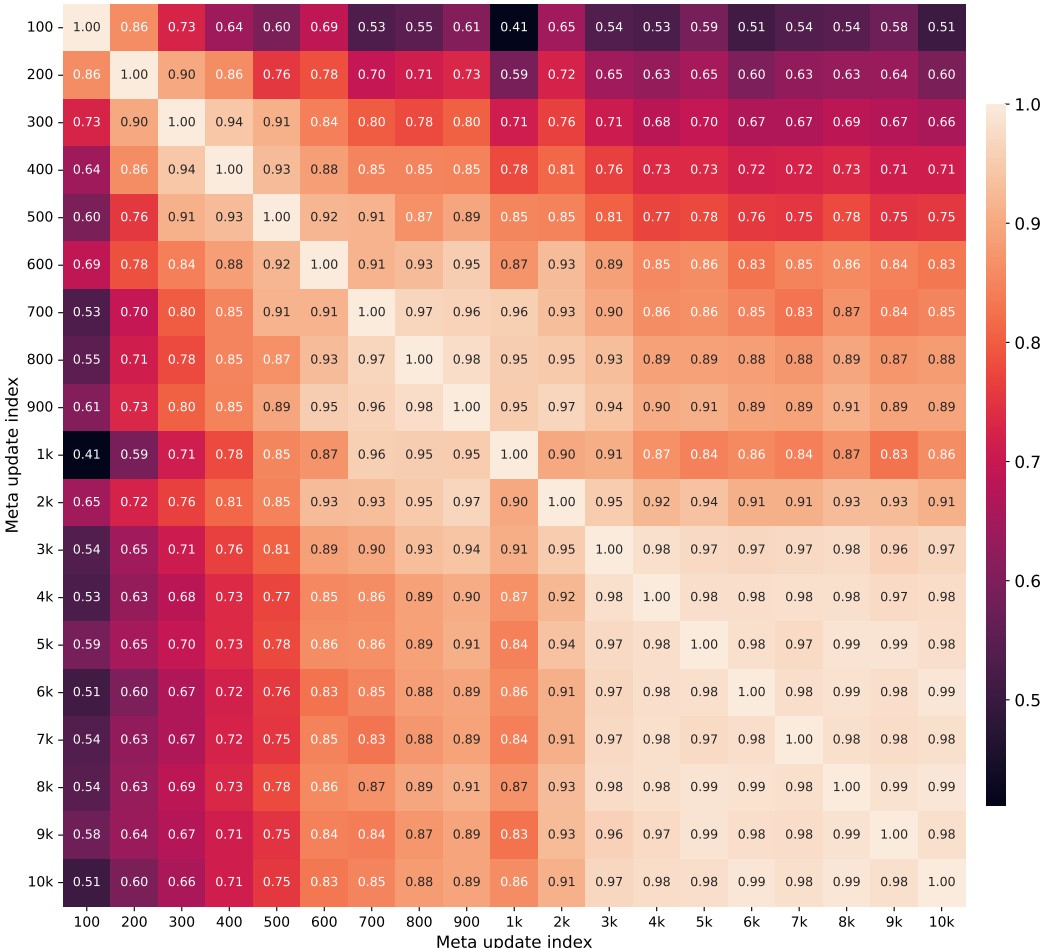

Figure 13: **Correlation of DataRater scores between different stages of meta-training.** This heatmap displays the Spearman correlation between DR scores (of the `Pile` 150M DR) obtained at various meta update checkpoints throughout training. All correlations are calculated using a static set of 25600 examples from the `Pile`. The plot demonstrates that the DR scores stabilize after a few thousand meta-training steps. While scores from early checkpoints show weaker correlation with later ones (as well as each other), scores from checkpoints after approximately 3000 steps become almost perfectly correlated (correlation $> 0.95$), indicating convergence.

ing heuristic features. We first computed the Pearson correlation between the DataRater score and all the heuristics which is shown in Figure 14. This analysis shows that the DataRater score is most strongly correlated with features related to document size, such as the number of packed subsequences, characters, words, and sentences. Conversely, the score is anti-correlated with measures of non-standard content, such as the fractions of non-alphanumeric characters and punctuation. Notably, the DataRater score has only a weak correlation ($0.22$) with the composite C4 filter, `c4_passes_all`, which aggregates all individual `C4` checks.

A simple linear regression model predicting the DataRater score from these features achieves a high $R^2$ of $0.766$. However, as many heuristics are highly collinear (i.e., strongly correlated or anti-correlated), this model is unsuitable for identifying the most predictive features. To isolate a minimal set of predictors, we instead fit a Lasso ($L_1$) regression model. We applied strong regularisation ($\alpha = 0.03$) and normalised all features (Z-scoring). We trained the Lasso model with 10-fold cross-validation (repeated 5 times), achieving an $R^2$ of $0.753$. The resulting coefficients, shown in Figure 15, reveal that the Lasso model set most feature weights to zero, identifying only a few as significant predictors. The score is most strongly and positively predicted by the number of packed sub-sequences (`num_articles`, $+0.42$). The presence of curly braces (`c4_curly_brace`, $+0.27$)

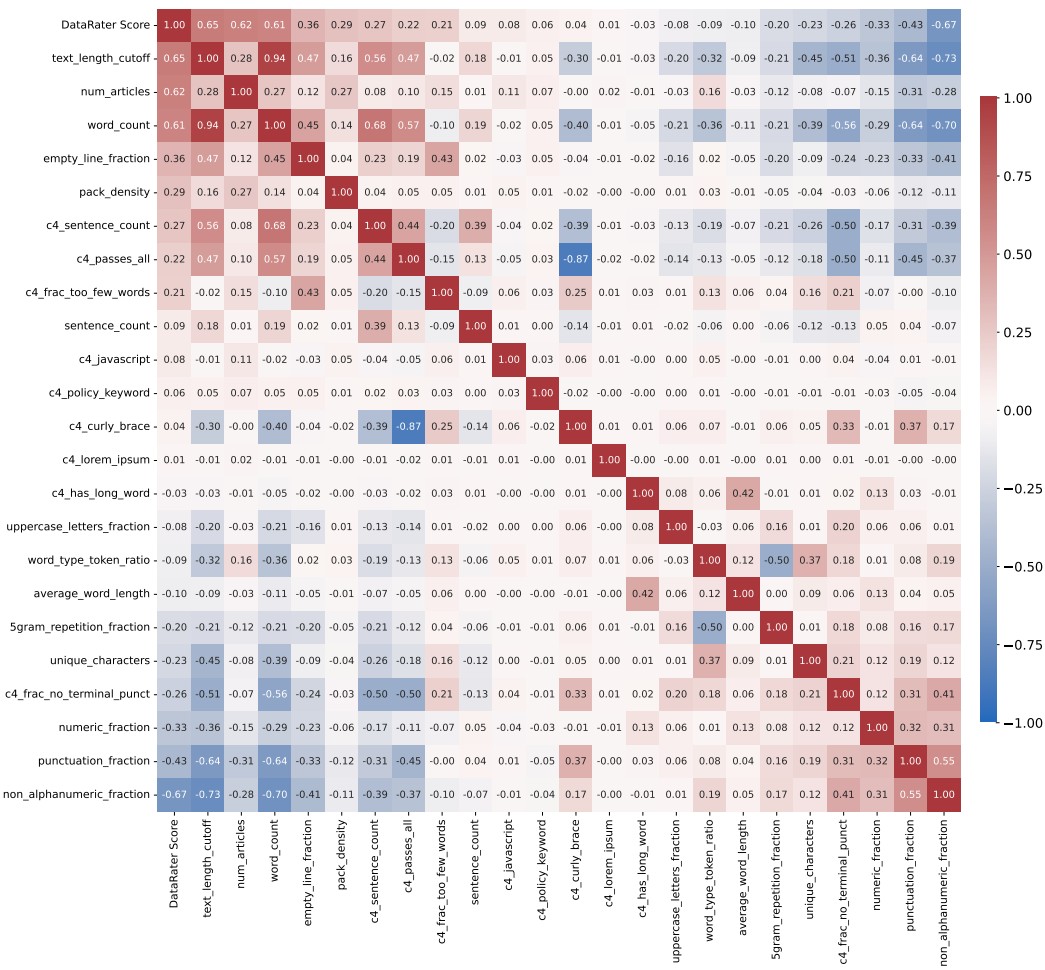

Figure 14: **Pearson correlation between DataRater score and various text filtering heuristics.** The score is most strongly correlated with document size metrics and anti-correlated with non-standard content, such as non-alphanumeric text and punctuation. Notably, the correlation with the composite C4 filter, `c4_passes_all`, is low (0.20).

and greater text length (`text_length_cutoff`, $+0.19$; `word_count`, $+0.13$) also yield moderate positive coefficients. The only strong negative predictor is a high fraction of non-alphanumeric characters ($-0.33$). In summary, the Lasso model indicates the DataRater score is primarily a function of document count (i.e., packed articles) and length, penalised by non-alphanumeric content. The Lasso model fit indicates that DataRater score is notably insensitive to most other common linguistic heuristics or C4-style filters.

**Experimental details.** For evaluation, we train causal auto-regressive LLMs from random initialisation, following the Chincilla model architecture, on data filtered by a DataRater model. We use the Adam optimiser [Kingma and Ba, 2015] with a learning rate of $0.001$ for LLMs of size 50M, 150M and 400M and one of $0.0002$ for 1B models, with a standard cosine learning rate decay schedule. We train 50M, 150M, 400M and 1B models for 5k, 12k, 30k and 48k update steps respectively, with a 128 batch size (except for 1B for which we use a doubled batch size of 256), with a sequence length of 2048. Token budgets per model size can be inferred from the number of update steps, batch size and sequence length. For meta-training, for each inner LLM, we use a batch size of 128 to compute the inner loss, and an outer batch size of 128 to compute the outer loss. We used the Adam optimiser, with a decoupled weight decay of $0.1$ [Loshchilov and Hutter, 2019], with 100 steps of linear warmup and global norm clipping [Pascanu et al., 2013] of $0.01$.

**Infrastructure details.** We implemented our infrastructure using the `jax` framework [Bradbury et al., 2018] and its ecosystem of associated libraries [DeepMind et al., 2020]. Our meta-training framework

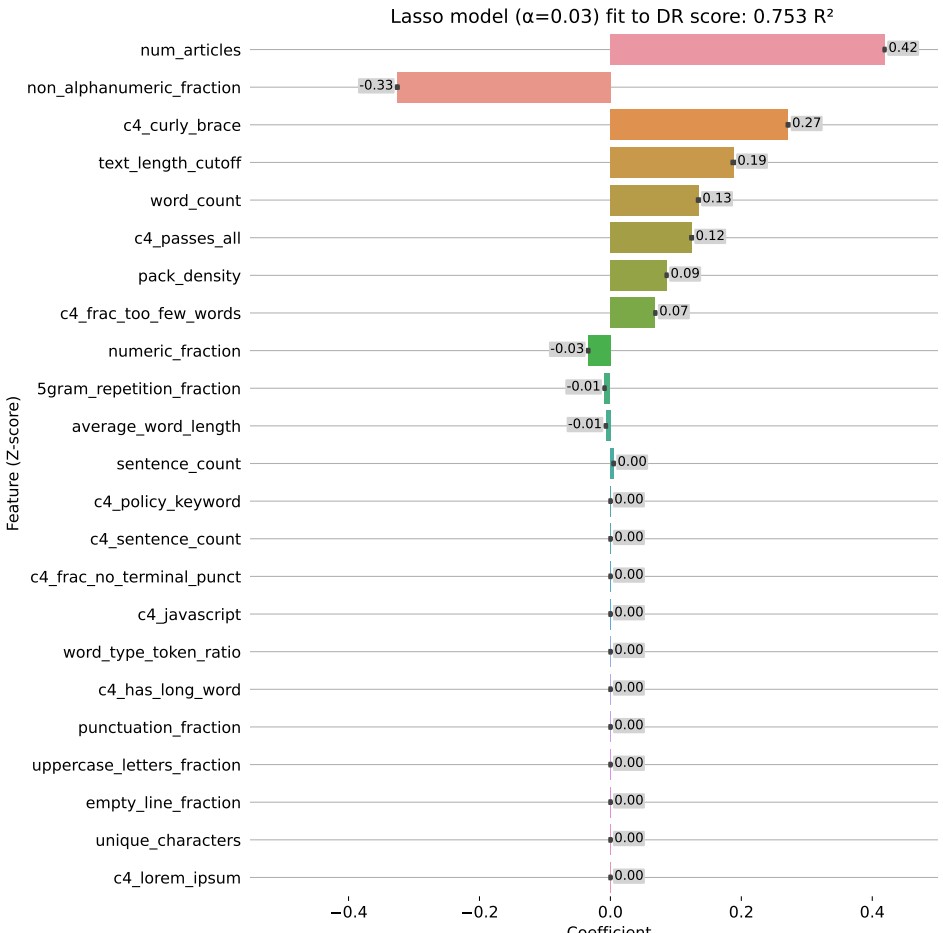

Figure 15: **Coefficients of a Lasso ($L_1$) regression model fit to predict the DataRater score from Z-scored heuristic features.** The model uses strong regularisation ($\alpha = 0.03$) and achieves an $R^2$ of $0.753$ (vs. $0.766$ of OLS) assigning most features a zero coefficient. The DR score is most strongly predicted by the number of packed sub-sequences, the presence of curly braces and features related to the document length. The only strong negative predictor is the proportion of non-alphanumeric content.

is inspired by the Podracer architecture for distributed reinforcement learning [Hessel et al., 2021], and it enables parallel computation of meta-gradients from the inner LLM population. We have run our experiments on Google TPUs: we used $4 \times 4 \times 4$ topology v5 TPUs for meta-training, and up to $4 \times 8$ topology v6e TPUs for evaluation. We carefully optimised our infrastructure, using data parallelism, model sharding as well as by using most techniques from MixFlow-MG [Kemaev et al., 2025] to keep RAM (i.e. HBM) consumption within hardware constraints.

**Discussion on DataRater framework extensions.** We dedicated the core paper to present evaluations on the most widely applicable instantiation of the DataRater framework, focusing on accelerating learning progress on a given and fixed pre-training dataset. This has clear practical benefits and, as we have shown, can result in significant computational efficiency gains.

In other experiments, we investigated online adaptation during training using dynamic data selection: i.e., we provided additional features (similar to and derived from RHO-LOSS [Mindermann et al., 2022]( which depend on the inner LLM state, as inputs to the DataRater model – here, results showed that the DataRater weights (aggregated at the coarse mixture level) become a function of the inner LLM training progress.

We also experimented with different loss weighting granularities: at the level of individual tokens (as the weighting used by Hu et al. [2023]) and at the coarse-grained mixture level (as the weighting used by Xie et al. [2023]). For our training efficiency meta-objective, we still obtained the best performance

by doing sequence-level filtering (vs. token-level weighting or coarse-mixture weighting). We also experimented with different downstream metrics (e.g., using NLL on English Wikipedia to meta-learn to identify high quality English data for pre-training) as well as non-differentiable meta-objectives (i.e., accuracy metrics, where we used Evolutionary Strategies [Rechenberg, 1973, Salimans et al., 2017] to estimate meta-gradients).

Finally, we also experimented with tailoring the datastream to narrowly targeted preferences by meta-learning a DataRater for curating instruction tuning data for the supervised finetuning regime; in this case we used set of human curated conversational data for the meta-objective, and this resulted in conversational data (e.g. forums, Quora, podcast data) being heavily upweighted by the DataRater.

## B    What can we expect from the DataRater?

To analyse the DataRater in a simple setting, consider the case when the test dataset $\mathcal{D}_{\text{test}}$ consists of clean samples only, while the training set $\mathcal{D}_{\text{train}} = \mathcal{D}_{\text{test}} \cup \mathcal{D}_{\text{corrupted}}$ contains a mix of clean data and fully corrupted data. Assume that the inner and outer loss functions are the same, that is, $\ell = L$ and are both next token loss.

In this case, if we assume that the corrupted data is completely useless, then we want the ratings assigned by the DataRater to clean data to be much larger than the ones assigned to corrupted data (i.e. $\phi_\eta(\mathbf{x}) \gg \phi_\eta(\mathbf{x}')$ for every $\mathbf{x} \in \mathcal{D}_{\text{test}}$ and $\mathbf{x}' \in \mathcal{D}_{\text{corrupted}}$), so that the effective final weight for each corrupted sample will be close to zero.

However, if the noise in the corrupted samples is much more benign, that is, the clean data has a lower noise level, then we would expect that it would still be valuable to use this corrupted data for training. Of course, with more noise, the smaller the DataRater weight should be – such a scenario is illustrated in Figure 3 of the main paper.

To understand this scenario better, consider the case of an overparametrised model in the interpolation regime, that is, where the loss for all clean data points is 0, i.e. $\ell(\mathbf{x}; \theta_T) = 0$ for all $\mathbf{x} \in \mathcal{D}_{\text{test}}$.

Then the outer loss over the whole dataset (2) (defined in the main paper) can be replaced with an arbitrary weighted combination of losses, with positive weights $w(\mathbf{x})$, where the goal is to minimise $L_w(\theta; \mathcal{D}_{\text{test}}) = \mathbb{E}_{\mathbf{x} \sim \mathcal{D}_{\text{test}}} [w(\mathbf{x}) \cdot \ell(\mathbf{x}; \theta_T(\mathcal{D}_{\text{train}}))]$.

Assuming each data point $\mathbf{x}' \in \mathcal{D}_{\text{train}}$ is a possibly noisy version of a test point $\mathbf{x} \in \mathcal{D}_{\text{test}}$ such that the gradient has the form $\nabla_\theta \ell(\mathbf{x}'; \theta) = \nabla_\theta \ell(\mathbf{x}; \theta) + \epsilon(\mathbf{x})$ where $\epsilon(\mathbf{x})$ is some zero-mean independent noise, then the training data results in an unbiased gradient estimate: $\nabla_\theta \ell(\mathbf{x}; \theta) = \mathbb{E}_{\epsilon(\mathbf{x})}[\nabla_\theta \ell(\mathbf{x}'; \theta)]$ with variance $\mathbb{E}_{\epsilon(\mathbf{x})} [\|\epsilon(\mathbf{x})\|^2]$. Thus, to have each sample contribute a gradient with uniform variance, one could choose $w(\mathbf{x}') = 1/\sqrt{\mathbb{E}_{\epsilon(\mathbf{x})} [\|\epsilon(\mathbf{x})\|^2]}$.

## C    Limitations

While the DataRater framework demonstrates promising results in enhancing training efficiency through meta-learned dataset curation, several limitations should be acknowledged:

**Sensitivity to Meta-Objective.** The effectiveness of the DataRater is inherently tied to the choice of the meta-objective (i.e., the outer loss and the held-out test data distribution) which defines the data valuation policy that will be encoded by the DataRater upon meta-training.

The DataRater framework itself explicitly allows for arbitrary inner and outer losses, and, in our *empirical evaluation* we chose the most widely applicable setting, of curating the initial dataset – i.e., where the meta-objective encodes improvement in training efficiency on the given training dataset.

However, the selection of appropriate meta-objectives which align with diverse end-goals remains a critical consideration; i.e., the learned data preferences might not generalise fully to significantly different downstream tasks or data modalities if the chosen held-out data for the meta-objective does not adequately represent these.

**Potential for Bias Amplification.** As discussed above, the DataRater learns to value data based on its provided meta-objective. If this objective, or the held-out data used to define it, contains biases,

the DataRater could inadvertently learn to amplify these biases – by upweighting data that aligns with them and down-weighting data that does not.

For example, if the held-out data underrepresents certain demographics or perspectives, a DataRater model might learn to devalue data pertinent to those groups, leading to models trained on the curated dataset that are less fair or equitable.

While our experiments focus on training efficiency with respect to a given data distribution, the framework's flexibility to use arbitrary outer losses also means it could be misused to optimise for undesirable model capabilities (i.e., if the meta-objective is chosen maliciously). Prior to releasing a curated dataset, careful consideration and auditing of the meta-objective (and the resulting data) are crucial to mitigate such negative societal impacts.

**Scalability of Meta-Gradient Computation.** The meta-gradient computation, while made feasible by techniques like MixFlow-MG [Kemaev et al., 2025], still presents computational challenges, especially as model scales continue to grow. Back-propagating through multiple inner model updates involves second-order derivatives and can, depending on the scale and number of updates involved, be resource-intensive in terms of both computation and memory (HBM).

We demonstrated upward scale generalisation: from 400M parameter inner models to 1B models for evaluation (which are trained on DataRater-filtered data, not used as inner models themselves) – i.e., showing generalisation to LLMs which require an order of magnitude more FLOPS to train. As discussed above, we have also experimented with scaling up meta-gradient computation for larger models (i.e., using inner models with 27B parameters) for the supervised fine-tuning regime, where inner model updates are made in a low-rank projection of parameter space using LORA [Hu et al., 2022]. This setting was non-trivial to tune for staying within hardware limitations on HBM consumption, but feasible. However, the scalability of meta-training DataRater models for extremely large foundation models with *fully dense* inner updates remains an open question, and may require further algorithmic advancements in scalable bilevel optimisation.

Table 1: Description of text filtering heuristics use for correlation analysis.

| Heuristic | Description |
|---|---|
| packing_density | Ratio of valid tokens to total tokens. |
| num_articles | Number of packed sub-sequences. |
| text_length_cutoff | Number of characters. |
| word_count | Total word count (whitespace-delimited). |
| sentence_count | Count of terminal punctuation (regex [.!?]). |
| empty_line_fraction | Ratio of empty lines (regex \n\s*\n) to total newline characters. |
| unique_characters | Ratio of unique characters to total text length. |
| word_type_token_ratio | Word-level type-token ratio, i.e. number of unique words to number of words. |
| non_alphanumeric_fraction | Fraction of non-alphanumeric characters (regex \W). |
| uppercase_letters_fraction | Fraction of uppercase letters (A-Z). |
| punctuation_fraction | Fraction of punctuation characters (regex [^\w\s]). |
| average_word_length | Mean word length (total non-whitespace chars / word_count). |
| numeric_fraction | Fraction of numeric digits (0-9). |
| 5gram_repetition_fraction | Fraction of duplicate (non-unique) word-level 5-grams. |
| c4_lorem_ipsum | True if the text contains "lorem ipsum". |
| c4_curly_brace | True if the text contains a curly brace ({). |
| c4_contains_javascript | True if any line contains the keyword "javascript". |
| c4_has_long_word | True if any space-delimited word exceeds a length of 1000. |
| c4_fraction_lines_fail_terminal_punct | Fraction of lines not ending with standard terminal punctuation (.?!"'). |
| c4_fraction_lines_fail_min_words | Fraction of lines containing fewer than 3 words. |
| c4_contains_policy_keyword | True if the text contains policy keywords (e.g., "privacy policy", "terms of use"). |
| c4_sentence_count | Sentence count derived from NLTK's Punkt sentence tokenizer, assuming English text. |
| passes_all_c4_filters | Composite filter: applies document-level (lorem ipsum, {}) and line-level (length, punctuation, keywords) filtering, then checks for a minimum of five sentences. |

| | | NLL (↓) | | | | Accuracy in % (↑) | | | | | | | | | | |
| | | Wikipedia | | Validation set | | HellaSwag | | SIQA | | PIQA | | ARC Easy | | CommonsenseQA | |
| | | Baseline | DR | Baseline | DR | Baseline | DR | Baseline | DR | Baseline | DR | Baseline | DR | Baseline | DR |
| Train Dataset | Model | | | | | | | | | | | | | | |
| The Pile | 50M | 3.66 | 3.30 | 3.17 | 2.94 | 26.01 | 26.47 | 39.41 | 40.07 | 55.22 | 59.36 | 28.25 | 31.58 | 22.19 | 23.51 |
| | 150M | 3.09 | 2.81 | 2.69 | 2.52 | 27.20 | 28.96 | 40.33 | 42.27 | 59.30 | 61.15 | 34.39 | 37.02 | 26.04 | 27.19 |
| | 400M | 2.69 | 2.52 | 2.36 | 2.27 | 31.59 | 34.71 | 42.43 | 43.91 | 62.62 | 65.02 | 40.35 | 40.53 | 29.65 | 28.17 |
| | 1B | 2.35 | 2.24 | 2.08 | 2.03 | 41.47 | 46.15 | 44.68 | 44.47 | 68.12 | 69.70 | 43.16 | 44.91 | 30.71 | 33.09 |
| C4/noclean | 50M | 4.17 | 3.94 | 2.93 | 2.69 | 26.05 | 26.51 | 39.87 | 39.46 | 54.90 | 57.34 | 28.07 | 31.23 | 22.85 | 22.85 |
| | 150M | 3.69 | 3.48 | 2.44 | 2.22 | 26.42 | 27.65 | 40.02 | 42.17 | 57.51 | 61.37 | 30.88 | 32.81 | 23.10 | 23.26 |
| | 400M | 3.35 | 3.24 | 2.04 | 1.91 | 29.52 | 31.28 | 40.74 | 42.12 | 61.32 | 63.44 | 32.28 | 34.74 | 24.57 | 26.62 |
| | 1B | 3.01 | 2.99 | 1.64 | 1.59 | 34.97 | 36.90 | 43.65 | 43.86 | 64.96 | 66.65 | 38.25 | 35.96 | 27.60 | 28.42 |
| C4 | 50M | 3.91 | 3.91 | 3.65 | 3.62 | 26.76 | 26.93 | 41.50 | 40.69 | 59.19 | 60.77 | 32.11 | 32.28 | 25.96 | 25.06 |
| | 150M | 3.45 | 3.45 | 3.20 | 3.18 | 30.42 | 30.95 | 43.91 | 43.24 | 63.28 | 64.36 | 37.02 | 38.95 | 27.35 | 27.52 |
| | 400M | 3.20 | 3.22 | 2.92 | 2.91 | 37.95 | 38.91 | 44.93 | 43.91 | 68.66 | 69.59 | 40.18 | 41.58 | 29.16 | 29.98 |
| | 1B | 2.95 | 2.97 | 2.64 | 2.63 | 50.32 | 50.81 | 44.17 | 45.65 | 72.47 | 72.63 | 42.11 | 43.16 | 31.20 | 32.02 |

Table 2: **Performance metrics** (as NLL or accuracy) at convergence across three train datasets, four model sizes and seven evaluation tasks (two negative log likelihood tasks, and five downstream accuracy tasks). "DR" stands for DataRater.

