# OpenReview forum: "DataRater: Meta-Learned Dataset Curation"
_NeurIPS.cc/2025/Conference — NeurIPS 2025 poster_

### Official Review · Reviewer_VwDw · 2025-06-27

**Clarity:** 3
**Significance:** 2
**Originality:** 3
**Rating:** 4
**Confidence:** 4

**Summary:**

The paper introduces a novel meta-learning approach called DataRater for curating data for training LLMs. The DataRater framework trains a data-selector model in a meta-learning fashion by optimizing through unrolled inner optimizations of an actual LM being trained. The paper conducts experiments across different model scales on three publicly available datasets. Empirical evidence shows that the trained DataRater models can select high-quality training data for LMs, thereby improving compute efficiency of training. Further analysis on the type of training data samples selected by the DataRater are also provided to guide intuitions on how DataRater works.

**Questions:**

- Doesn’t DataRater require manual tuning of the threshold as well? It seems a bit disingenuous to claim that your method does not need manual tuning (in line )? It requires the same amount of tuning as a fast-text classifier pretty much?
- Does DataRater also work for >1-epoch training or is it limited to the single-repeat setting? From the description, it is unclear if the experiments are conducted for a single or multi-epoch setting (especially given the different discard fractions used).
- Why do the inner model parameters need to be reset? Does this not making the DataRater somewhat off-policy, compared to the learning signal gotten from the updated inner models? A more detailed explanation of this would be good to add in the paper.
- One thing that is not clear to me from this statement in lines 168-172: Is this using offline DataRater or an online version that filters out every batch? Do you assume amortization of running the DataRater model on the full dataset once, or do you still count that towards the total FLOPs needed for training? The claims in the paper around compute efficiency sometimes use number of training steps, and sometimes use FLOPs. If indeed the cost of running inference of DataRater is not accounted for, making claims about compute efficiency purely using number of training steps is not fair in my opinion.

**Ethical Concerns:**

["NO or VERY MINOR ethics concerns only"]

**Final Justification:**

After the rebuttal, most of my concerns have been addressed, and so I am slightly leaning towards acceptance.

**Limitations:**

Yes, the paper provides a good list of limitations, both technical and societal, in the appendix.

**Quality:**

2

**Strengths And Weaknesses:**

Strengths:
- The paper is very well written and provides a great overview of the method, both from an intuition perspective and a technical implementation perspective.
- The problem outlined by the paper---improving training efficiency via data curation---is an extremely important problem and the paper does a good job of characterizing it.
- The intuitions provided by the analysis in Section 3 (Figures 7 and 8) are very helpful in understanding what data samples are selected by DataRater. More papers in the data curation literature should adopt this.

Weaknesses:
- From figure 1 and as discussed in Section 3, it seems like DataRater brings large boosts for relatively under-filtered datasets like Pile and C4/no-clean, whereas for a reasonably well filtered dataset like C4, the gains seem to be a little bit more modest. This raises the question if DataRater is complementary to other LM data curation / filtering methods or not? The benefits of the SoTA filtering pipelines used in current LMs is that they are, for the most part, complementary i.e. you can use multiple fast-text classifiers together, and performance improves (see [Nemotron-CC](https://arxiv.org/abs/2412.02595) paper). Hence, it is an important question to answer if the DataRater method is complementary or orthogonal to more such filters i.e. can DataRater be used as one filter in a stack of many more filters?
- DataRater assumes access to a test distribution. In the set of experiments conducted, that is just from the same distribution as the training distribution. What happens when your test distribution is quite different from your training distribution? For example, could the authors discuss / ground their findings with respect to the experiments conducted in the [Color-filter](https://proceedings.neurips.cc/paper_files/paper/2024/file/b0f25f0a63cc544d506e4c1374a3c807-Paper-Conference.pdf) did, where they explicitly have a “validation set” that is the target distribution of interest.
- The paper misses out on a lot of relevant literature that also tackles the same problem of data curation for LMs. Some notable omissions are: [DataComp-LM](https://arxiv.org/abs/2502.10341), [AskLLM](https://arxiv.org/pdf/2402.09668), [Qurating](https://arxiv.org/abs/2402.09739), [Nemotron-CC](https://arxiv.org/abs/2412.02595), [DSIR](https://arxiv.org/abs/2302.03169), [WebOrganizer](https://arxiv.org/abs/2502.10341), [SemDeDup](https://arxiv.org/abs/2303.09540) etc.
- There is a lack of good baselines. All the experiments conducted only show comparisons to a standard baseline that is trained on uniformly sampled data. However, simple baselines like training a fast-text classifier on the same training data distribution as DataRater as the positives are missing. It is unclear if DataRater would substantially outperform such simple data selection methods, justifying the additional cost of the meta-learning training (given that training a fast-text classifier is extremely cheap in comparison).

---

> ### Author Rebuttal · Authors · 2025-07-31
>
> We are grateful the reviewer found our paper very well written, the problem statement important, and our qualitative results insightful.
>
> Please find our replies below.
>
> ***Tuning of DR***
>
> The DataRater does not need a labeled golden high-quality dataset, so in this sense requires significantly less curation effort than building a fast-text classifier. The DR filtering threshold (i.e. discard proportion) can be tuned at the smallest model scale, as shown in Figure 4, as this is shown to generalize upwards to larger model scales.
>
> ***Multi-epoch setting***
>
> We can train the DR in a single or multi-epoch setting, and found it works well in both. However, in our presented version, the DR does not explicitly account for repeats since this information is not provided as an input to the DR model. A more contextually-aware DR, which would include model-state dependent info as DR inputs, could be an interesting avenue for future work.
>
> ***Why reset inner model parameters***
>
> We wish to use a frozen trained DR to filter datasets for future downstream training runs. For this reason, we want it to generalise across all stages of training, rather than fit to a particular model at a particular stage of training. We found that having a population of inner models with resets to maintain diversity over the stage of training helps to promote this generalisation. As suggested, we will expand on this point in the final paper version.
>
> ***Offline vs Online, Amortization***
>
> > One thing that is not clear to me from this statement in lines 168-172: Is this using offline DataRater or an online version that filters out every batch?
>
> This is using an online DataRater version to filter data batches online. The overhead incurred by the DR inference cost is shown in pink in Figure 1.
>
> > Do you assume amortization of running the DataRater model on the full dataset once, or do you still count that towards the total FLOPs needed for training?
>
> We do not assume DataRater inference amortization. We count its inference costs in the total FLOPs needed for training.
>
> > The claims in the paper around compute efficiency sometimes use number of training steps, and sometimes use FLOPs. If indeed the cost of running inference of DataRater is not accounted for, making claims about compute efficiency purely using number of training steps is not fair in my opinion.
>
> In Figure 1, we explicitly account for the cost of running inference to filter data online using the DataRater model: the compute overhead (as a proportion of FLOPS) due to the inference cost of the DataRater is shown in pink. To calculate this we converted both the LLM training step cost into FLOPs as well as the DR inference step cost into FLOPs (while accounting for batch-size oversampling prior to DR online filtering). As the reviewer points out this is a critical point for the fairness of our claims, and we will update lines 169-170 as well as the caption of Figure 1 to make this accounting unambiguous.
>
> ***Complementary of filtering pipelines***
>
> All datasets we use have various degrees of heuristic filtering applied to them (e.g. for the Pile see Appendix C in [1], for C4 see page 6 of [2]).
>
> Our quantitative results show that DR filtering is complementary to the filtering done in the Pile, as well to the one in C4/en.noclean, with less complementarity over the filtering used in C4/en.
>
> Our qualitative findings indicate that the DR tends to pick up on unusual text (poorly encoded, OCR errors, non-standard characters) and highly-patterned uninformative text (long lists, tables, indices, log outputs), which can be complementary to various heuristic filters. The fact that the DataRater still finds gains on top of these strong pre-filtered datasets is, in our view, a key strength of the method.
>
> [1] Gao et al., The Pile: An 800gb 314 dataset of diverse text for language modeling.
>
> [2] Raffel, Shazeer, Roberts, Lee et al., Exploring the limits of transfer learning with a unified text-to-text transformer.
>
> ***Test distribution choices***
>
> We have experimented with applying the DR framework to test distributions different from the training distribution (Appendix A). E.g., when using English Wikipedia as the test distribution, the resulting DR downweighted user forum data and foreign language text while upweighting data from the most common websites. Similarly, when using a dataset of code examples as the test distribution, the resulting DR upweighted scientific and mathematical text.
>
> This use case requires selecting downstream metrics a priori which is often hard to reconcile with the breadth of capabilities required by foundational model training. So we did not regard this use case as the most practical one, and instead we focused our paper on the general case of self-supervised data curation.
>
> ***Additional data curation literature***
>
> We thank the reviewer for their comprehensive suggestions and will include the additional references provided in the final version of the paper.
>
> ***Fast-text classifier***
>
> A fast-text classifier needs labeled data in the form of, e.g., positive and negative examples. In the case where such labelled data is given, of course practitioners should experiment with a fast-text classifier. However, our method tackles the more general case of self-supervision, where no golden set is given.
>
> We hope our replies have suitably addressed your concerns. Please let us know if you have any other questions.

---

> > ### Comment · Reviewer_VwDw · 2025-08-07
> > **Response**
> >
> > Thanks to the authors for the detailed responses. Most of my concerns are addressed. However, the last point about insufficient baselines is still not addressed. Do the authors plan to include any additional baselines, or if they did already, could they please point me to those results?
> >
> > I think that is an important point to clarify which has also been raised by other reviewers (pnjF, JMg5 and i2Zt).

---

> > > ### Author Response · Authors · 2025-08-07
> > >
> > > Thank you for your valuable comments. We agree that including a naive baseline would strengthen our paper and we will add one to the final version.
> > >
> > > Our proposed baseline will be a perplexity-based filtering approach; specifically, we will:
> > > - Train a language model on the pre-training dataset (e.g., the Pile).
> > > - Use this model to score each batch of data and filter out those with the highest perplexity.
> > >
> > > This method is an adequate baseline because, critically, it does not use a golden validation set. By deriving its filtering signal from the pre-training data alone, it provides a like-for-like comparison to our own method.

---

### Official Review · Reviewer_pnjF · 2025-07-01

**Clarity:** 3
**Significance:** 3
**Originality:** 2
**Rating:** 4
**Confidence:** 3

**Summary:**

The paper "DataRater: Meta-Learned Dataset Curation" proposes a method for estimating the value of individual data points for training foundation models, using meta-learning with meta-gradients. The authors frame their contribution as making LLM training more data-efficient, especially since we are running out of human-curated text to train on.

**Questions:**

* How does DataRater compare against recently learned data selection methods?

* Since the DataRater is trained to optimize performance on a held-out subset of the same dataset, how do you ensure that it isn’t overfitting to the meta-validation metric or gaming spurious correlations in that specific split? Could you explore robustness to alternative outer losses/distribution shift between D_train and D_test/generalization to truly different downstream tasks?

* On a similar note, how sensitive is DataRater to the choice of outer loss, or shifts in domain/task?

Nitpicking: Please make sure that your figures are in the right place and that you refer to them in the text itself (e.g., the first time you refer to figure 1, which is on page 2, is on page 6; figures 2-4 were never referred to).

**Ethical Concerns:**

["NO or VERY MINOR ethics concerns only"]

**Final Justification:**

I am keeping my score as it captures the scope (that excludes downstream tasks) and contributions since the authors addressed the baseline issue, I am not reducing my score.

**Limitations:**

yes

**Quality:**

3

**Strengths And Weaknesses:**

**Strengths:**

* Topic: The problem of curating training data for foundation models, especially with increasing reliance on synthetic and noisy data, is critical and underexplored at scale.

* Empirical results: Consistent performance and compute savings across model scales (50M to 1B) and diverse datasets (The Pile, C4, C4/noclean). Also, the authors show that a DataRater trained on 400M inner models generalizes to smaller and larger models. This is practically useful and rarely demonstrated in other works. Cute qualitative analysis.

* Theoretical grounding: Meta-learning data value, bilevel optimization, and filtering via data scoring are all well-explored. The novelty here is in their combination. This is a rather incremental novelty, but still a novelty.


**Weaknesses:**

* Baseline: There’s no head-to-head comparison with existing learned filtering approaches like SEAL, JST, or PreSelect (only to the unfiltered baseline). There's also no vanilla baseline comparison, as I'd expect in figure 4.

* Downstream utility: The evaluation focuses on proxy tasks (HellaSwag, PIQA), not on a large downstream deployment (e.g., instruction tuning, chat models). Claims about general usability or production utility would benefit from such an example.

* Compute cost: Training the DataRater takes ~58% the compute of training a 1B LLM. The argument about amortizing this cost is valid, but practically speculative without end-to-end downstream deployment scenarios.

* High-quality data: The method is much more effective on noisy data (Pile, C4/noclean); on already filtered datasets like C4, the benefits are small and sometimes ambiguous.

* Technical details: While referencing MixFlow-MG and other tricks for scalability, the paper provides limited detail on hyperparameters, stability, or how these techniques differ from baseline approaches in practical terms. For reproducibility or for researchers hoping to build on this, more clarity would be welcome.

---

> ### Author Rebuttal · Authors · 2025-07-31
>
> We thank the reviewer for their comments; we are grateful they found our choice of problem domain important, results strong, and approach novel.
>
> Please find our replies below.
>
> ***Learned data selection methods***
>
> SEAL [1] targets the fine-tuning regime (i.e. adapting pre-trained LLMs to excel at specific downstream tasks), while our method targets the pre-training regime (i.e. training foundational models from scratch). SEAL builds a data selector for enhancing safety / lowering toxicity; our method learns to generally curate data. SEAL requires as input a safety-aligned LLM as well as a golden validation set of safe data points; our method does not require either. SEAL uses a bilevel optimization formulation as well, and approaches it using a penalty-based reformulation, instead of computing exact meta-gradients.
>
> JST [2] performs a two stage filtering process, which relies on a held out set of low-quality data for its predictions. This is in contrast to our work, where the validation data is sampled from the same distribution as the training set, as evidenced by all our results.
>
> PreSelect [3] identifies data points which correlate well with improvements in downstream metrics and shows that filtering those that do not result in performance improvements. Their method makes considerably stronger assumptions: it requires a set of downstream evaluation metrics (12 used, §2.3), as well as a held out validation set of high quality data (they select random samples from the top 3000 most frequent domains, like wikipedia.org, §2.3). Our method requires no curated validation set nor downstream metrics.
>
> To summarise, the recent learned data selection methods differ considerably from the DR in multiple significant dimensions, precluding meaningful direct comparisons. I.e., SEAL and the DR differ in the problem setting, goal and assumptions. The DR does not require validation data, while both JST and PreSelect do, while PreSelect also depends on having a set of downstream metrics as input.
>
> The points in Figure 4 with an x-axis value of 0% filter fraction correspond to no filtering, i.e. the raw dataset, which could be considered a default baseline.
>
> [1] Shen et al., SEAL: Safety-Enhanced Aligned LLM Fine-Tuning via Bilevel Data Selection
>
> [2] Zhang et al., Just Select Twice: Leveraging Low Quality Data to Improve Data Selection
>
> [3] Shum, Huang et al., Predictive Data Selection: The Data that predicts is the Data that teaches
>
> ***Overfitting to meta-validation metrics***
>
> We use strictly held-out validation sets as well as downstream evaluations in order to guard against meta-overfitting. In fact, we find in our setting this is not an issue. Since we use a split of the original (large) dataset, it is quite easy to have a diverse enough meta-validation set to prevent overfitting.
>
> ***Different downstream tasks***
>
> We have also tried using targeted meta-validation sets with a significant distribution shift from the original dataset. This also works quite effectively, although meta-overfitting is a greater risk. For this paper, we decided to focus on the matching-distribution setting, as we were excited by the generality of the approach which does not require any curation of a golden targeted/high-quality meta-validation dataset.
>
> ***Sensitivity to outer loss / domain / task***
>
> Overall we found the approach quite robust to the outer loss and choice of dataset. For example, preliminary experiments with code-focused datasets in the inner and/or outer loss showed similar qualitative results to those presented.
>
> ***Figures***
>
> Thank you for pointing this out. All figures are referenced but not in order. For the final paper version we will correct the Latex referencing so all figures are numbered in the order they appear in the PDF.
>
> ***Downstream utility***
>
> We use standard proxy tasks (HellaSwag, PIQA, etc.) as they are established benchmarks for assessing the quality of pre-trained foundational models at these scales. While evaluation on end-to-end instruction tuning or chat models is an interesting avenue for future work, the consistent improvements we observe across these diverse benchmarks (73 out of 84 configurations) strongly suggest the general utility of the DR.
>
> ***Compute cost***
>
> We acknowledge the upfront cost of training the DataRater which is ~58% of the FLOPS for a 1B LLM. This cost is rapidly amortized because the filtered LM datasets are typically reused. Given the demonstrated cross-scale generalization, the DR can be used for subsequent training of much larger models or multiple ablation studies, leading to significant net compute savings.
>
> ***Technical details***
>
> We provide more experimental details in the appendix, including on hyper-parameters (learning rate, model sizes,  optimizer, batch sizes, etc.), and infrastructure (Jax codebase, types of accelerators used for training, libraries, etc.). However, to improve clarity, for the final version we will expand this section to better detail how we make use of techniques from MixFlow-MG in our work.
> On the topic of meta-training stability, we employ several key techniques to stabilize meta-gradient computation, including: using a population of inner models (line 129); passing the meta-gradient for each inner model through an individual Adam optimizer (line 130) and averaging the resulting updates; and periodically reinitializing the inner models to stratify learning progress (lines 131-132).
>
> We hope our answers have suitably addressed your concerns. Please let us know if you have any other questions.

---

> > ### Comment · Reviewer_pnjF · 2025-08-04
> >
> > Thanks for addressing all of my comments.
> >
> > - Baselines: Is there any naive baseline that would make sense here? I am familiar with works that "invent the wheel" in the sense that they are very different from everything else out there, but most of them do try to adapt existing methods to serve as a proxy baseline, while acknowledging the differences.
> >
> > - Overfitting: Can you elaborate (preferably with numbers) on this claim? "In fact, we find in our setting this is not an issue."
> >
> > - Downstream tasks: Can you elaborate on this point? "This also works quite effectively, although meta-overfitting is a greater risk." Moreover, since you decided this is not part of the scope of this paper, the claims about general usability should be toned down.

---

> > > ### Author Response · Authors · 2025-08-06
> > >
> > > Please find our replies below:
> > >
> > > **Baselines:**
> > > An adequate naive baseline could be: filtering based on perplexity of an LLM trained on the same pre-training dataset. For example, a like-for-like setup could be: (1) train a scoring LLM on the Pile and (2) use the pre-trained scoring LLM to drop the data points from each batch which have the highest perplexities. This would be a very coarse proxy for quality (vs. using a golden set), but, for completeness, we will include this naive baseline in the final version.
> > >
> > > **Overfitting:**
> > > Of course. In the appendix, in Figure 9, we show that an aggregate metric of validation performance improvement remains relatively stable as DR meta-training progresses depending on hyper-parameters (i.e. specifically for DRs trained with 8x400M inner models and evaluated on a 400M model, the aggregate performance metric remains approximately stable at approximately 80% from meta iteration 2000 to meta iteration 8000).
> > >
> > >
> > > **Downstream tasks:**
> > > Of course, we are happy to elaborate: we meant that meta-overfitting is a greater risk when the size of the validation set is smaller (e.g., English Wikipedia is. ~20GB vs. C4/en which is ~300GB) and might not encompass the breadth of capabilities required for pre-training.
> > > Regarding general usability, thank you for your feedback; we will revise the text to clarify the claims that are directly in scope for this work and supported by our experiments.

---

### Official Review · Reviewer_JMg5 · 2025-07-02

**Clarity:** 3
**Significance:** 2
**Originality:** 3
**Rating:** 4
**Confidence:** 3

**Summary:**

This paper introduces DataRater, a meta-learning framework for automated dataset curation that learns to estimate the training value of individual data points for foundation models. The approach addresses the current reliance on manual heuristics and coarse-grained data mixing in large-scale model training.

DataRater uses a transformer-based scorer that assigns scores to data points, trained via bilevel optimization with meta-gradients. The meta-learning objective is to improve training efficiency on held-out validation data. The method employs a continuous relaxation of discrete data selection, using softmax-weighted gradients during training and top-K filtering during actual curation.

The authors evaluate DataRater across three datasets of varying quality (C4, C4/noclean, The Pile) and four model scales (50M to 1B parameters). A key finding is cross-scale generalization: a DataRater trained with 400M parameter inner models effectively transfers to target models ranging from 50M to 1B parameters, with consistent optimal filtering proportions across scales.

The method achieves substantial compute savings on lower-quality datasets, with up to 46.6% net compute reduction on The Pile while matching baseline performance. Results show DataRater learns to down-weight data that aligns with human intuitions of poor quality, such as OCR errors, encoding issues, and non-English content. The approach demonstrates improvements across 73 out of 84 experimental configurations spanning multiple datasets, model scales, and evaluation metrics.

The technical implementation leverages MixFlow-MG for scalable bilevel optimization, making the approach computationally feasible for realistic model and dataset sizes. The framework supports both batch-level filtering for efficiency and individual data point scoring for parallel processing pipelines.

**Questions:**

**1. Baseline Comparisons** How does DataRater compare against established curation methods beyond unfiltered data? Please provide comparisons against: (a) heuristic filtering similar to C4's pipeline applied to The Pile, (b) perplexity-based filtering, and (c) random subsampling at equivalent discard rates. **Criterion**: Adding proper baselines would increase Quality if DataRater shows meaningful advantages over ablated heuristics.

**2. Cost-Benefit Analysis and Decision Framework** When is DataRater worth the ~58.4% training cost versus cheaper alternatives? Please provide: (a) clear decision criteria for when to use DataRater vs. heuristics, (b) break-even analysis for different dataset quality levels, and (c) guidelines for practitioners. **Criterion**: Clear practical guidance would increase Significance.

**3. Systematic Analysis of Learned Patterns** Can you extract interpretable rules from DataRater beyond qualitative examples? Please provide: (a) quantitative correlation analysis between scores and measurable text features, (b) comparison of learned patterns vs. existing heuristics, and (c) feature importance analysis that could inform simpler filtering methods.

**Ethical Concerns:**

["NO or VERY MINOR ethics concerns only"]

**Final Justification:**

I keep a borderline-accept because the paper’s main strengths—a meta-learned data filter that delivers compute savings and robust cross-scale transfer—are technically solid and practically promising. The rebuttal usefully adds quantitative correlation evidence and pledges a fuller analysis of what the DataRater learns, partly addressing my interpretability concern, and provides qualitative guidance on amortising its 58 % meta-training cost. However, it still lacks empirical comparisons with heuristic, perplexity-based, or random filtering, so the key baseline gap and cost-benefit uncertainty remain. On balance, the contributions merit publication, but I urge the authors to include the missing baselines and clearer decision guidelines in the final version.

**Limitations:**

yes

**Quality:**

3

**Strengths And Weaknesses:**

### Strengths
- **Quality**: The technical approach is sound with proper bilevel optimization formulation and scalable implementation using MixFlow-MG. The cross-scale generalization finding (DataRater trained on 400M models works for 50M-1B targets) is empirically robust and theoretically interesting. Experiments span multiple datasets, model scales, and metrics, providing comprehensive evaluation.
- **Clarity**: The paper is well-written with clear mathematical formulation and algorithmic specification. The meta-learning framework is explained intuitively, and experimental setup is thoroughly described. Figures effectively communicate key results, particularly the compute efficiency gains.
- **Significance**: Addresses an important scalability challenge in foundation model training. Achieves substantial compute savings (up to 46.6%) on noisy datasets while maintaining performance. The cross-scale generalization property has important practical implications for amortizing curation costs across model sizes.

### Weaknesses
- **Quality**: gap in baseline comparisons—only compares against unfiltered data, not against established heuristic filtering, perplexity-based methods, or random subsampling. This makes it impossible to assess whether neural approaches justify their computational overhead versus simpler alternatives.
- **Clarity**: Limited analysis of what DataRater actually learned beyond qualitative examples. No systematic feature analysis or rule extraction that could provide transferable insights to the community.
**Significance**: Minimal gains on well-curated datasets (C4) where most practitioners operate, limiting immediate practical value. Training cost (~58.4% of target model) isn't justified by small improvements on quality data.
- **Originality**: While the application is novel, the core bilevel optimization techniques are established. The continuous relaxation approach follows standard practices in differentiable discrete optimization.

---

> ### Author Rebuttal · Authors · 2025-07-31
>
> We thank the reviewer for their extensive summary of our work; we are grateful they found our technical approach sound and scalable, our writing clear, and our problem statement practical.
>
> Please find our replies below.
>
> ***Baseline Comparisons***
>
> Our experiments, in particular on C4 and C4/en.noclean, give some context for (a) as to the comparison with strong heuristic filtering methods. Using the DR on the no-clean variant does not recover the performance of C4 using the fully tuned heuristic filtering pipeline. However, we can still improve over C4 on most metrics when using that as the starting point, albeit with diminishing returns.
>
> We think the DR approach is exciting because it requires so little prior knowledge, deriving a filtering strategy merely from the pre-training data itself. We consider this finding to be of significant scientific interest. It may be of significant value to practitioners as well. Clean data sources, especially for niche domains, are quickly becoming exhausted. Teams working on foundational models are constantly searching for new sources of data, which have increasingly poorer base quality and poorly understood patterns that may be used for heuristic filtering.
>
> Regarding random subsampling (c), we highlight that our method often improves the final performance (Figures 5 and 6) compared to training on the full, unfiltered dataset. Random subsampling reduces compute but typically does not improve final performance over the full dataset. The substantial gains we observe strongly indicate that the DR is performing non-trivial, quality-based selection superior to random chance, selectively removing data found to hinder training (e.g., OCR errors, encoding issues).
>
> Concerning perplexity-based filtering (b), these methods typically require a high-quality reference corpus or a well-trained reference model to establish scores. A key advantage of the DR is its self-supervised nature, learning value directly from the pre-training data without external references/validation sets. Furthermore, perplexity can be a coarse proxy for quality—potentially discarding complex yet valuable data—whereas DR optimizes directly for learning efficiency via the meta-objective.
>
> ***Decision Framework***
>
> It is difficult to provide a comprehensive decision framework as the break-even point depends significantly on the intended downstream use of the DR. If one does multiple training runs using the filtered data (e.g. for modeling ablations), or scales up a downstream training run as we show is possible, the cost of training the DR is rapidly amortized.
> For a dataset where existing heuristics have not been extensively tuned, and a practitioner expects to use an order of magnitude more compute in downstream use (for repeated future training runs or scaling up), we can recommend trying the DR strategy for filtering.
>
> ***Learned Patterns***
>
> We conducted a preliminary analysis of the DR, and found low (Pearson) correlation (< 0.15) with standard heuristics from the C4 pipeline like maximum word length, terminal punctuation checks, word repetition counts, curly brace checks. We found the DR tended to pick up on unusual text (poorly encoded, OCR errors, non-standard characters) and highly-patterned uninformative text (long lists, tables, indices, log outputs). Overall our analysis indicates that the DataRater is not a simple heuristic (since there’s not a single strong correlation to any of the heuristics we tested) and that most correlations are negative, i.e., the DataRater learns to penalize markers of low-quality text. We will include the full analysis in the final paper version.
>
> It would certainly be possible to construct heuristic filters to capture some of these features. It is also possible to distill the DR into a smaller and simpler model.
>
> We hope our answers have suitably addressed your concerns. Please let us know if you have any other questions.

---

### Official Review · Reviewer_i2Zt · 2025-07-02

**Clarity:** 3
**Significance:** 4
**Originality:** 4
**Rating:** 5
**Confidence:** 4

**Summary:**

The paper deals with finding most influencial training data for foundational models in order to improve training efficiency, but also possibly predictive performance. The goal of a filtering process is to select a subset of training data that leads to the smallest test error. The DataRater approach is to solve a continuous linear relaxation for discrete subsampling of data, where for each gradient update and minibatch a continuous weight in [0,1] is chosen. The weights are represented by a learned function, the DataRater model, where the outputs are normalized by a softmax preference function Parameter opt. is done via meta-Learning of DataRater model parameteres by backpropagrating through multiple inner loops. The authors build their implementation on prior work MixFlow-MG and a population of inner models for meta-gradient calculation, in order make optimization feasible and stable. The data curation is then performed using top-K filtering, i.e., removing data with low predicted weight.

The system is evaluated by learning DataRater, then training a LM of sizes up to 1B from random init. on the resulting data. Datasets used are C4 variants and Pile. The Inner LLMs and DataRater model are based on Chinchilla architectures. The final accuracy is measured on HellaSwag, SIQA, PIQA, ARC Easy, Commonsense QA downstream tasks. The results show that DataRater accelerates learning (i.e., FLOP savings) esp. for lower quality datasets. The authors also evaluate the best data drop rates, which correlates with the amount of available noise.

**Questions:**

- The pipeline first trained a DataRater model using multiple inner LLMs, then trains the final LLM using the curated data. Could one also train such a final LLM directly, similar to the inner LLMs? I understand that the DataRater model should be reused multiple times to actually save compute, but could end-to-end optimization also fine the optimal amount of needed data?
- Would Multi-criteria optimization be possible to also reduce dataset size directly?
- What are the advantages over competing approach mentioned in related works - e.g., influence functions, SHAP values, JST, ... ? More efficient? Better quality assessments?
- T=2 inner models updates have been found to be best. Did the performance remain stable with more updates and steadily increase? Can you please give possible explanations for the good fit of T=2?
- How complex are the chosen datasets? Do the findings generalize to any complexity of text-based downstream task?
- Is the optimization process sufficiently stable for updating a trained DataRater wrt new data?

**Ethical Concerns:**

["NO or VERY MINOR ethics concerns only"]

**Final Justification:**

I thank the authors for their answers to my questions. After reading the answers and the comments of the other reviewers, my assessment to accept the paper clearly remains. The proposed method to estimate the value of training data points is clearly described, sensible and provides reliable results. The proposition to add further baselines is of course welcomed.

**Limitations:**

Limitations become clearer when answering the questions in the above section.

**Quality:**

4

**Strengths And Weaknesses:**

## Pros
- Clear presentation of methods and results
- Sensible and original meta-learning approach to learn a data value function
- Good empirical results for the method, showing efficiency and predictive performance improvements compared to training with all data

## Cons
- No empirical comparison to competitors and not sufficient conceptual argumentation why new approach is adding value wrt existing work.
- Missing argumentation for the choice of datasets wrt (task-) complexity

---

> ### Author Rebuttal · Authors · 2025-07-31
>
> We thank the reviewer for their comprehensive summary of our work, and are grateful they found our paper's methods and results clear, our approach sensible, and our empirical results good.
>
> We provide individual replies to the reviewer's questions below:
>
> > The pipeline first trained a DataRater model using multiple inner LLMs, then trains the final LLM using the curated data. Could one also train such a final LLM directly, similar to the inner LLMs? I understand that the DataRater model should be reused multiple times to actually save compute, but could end-to-end optimization also fine the optimal amount of needed data?
>
> Yes, one could train a single DataRater model side-by-side with a single LLM whose data could be filtered online by the DR. This type of DR model could provide targeted data relevant for the current state of the LLM -- e.g., if performance on maths appears to lag, this could be rectified immediately with relevant maths data. We conducted a few experiments in this direction (see Appendix L951), but we leave a full treatment of this line of research to future work.
>
> In our case, indeed, as the reviewer points out, we save compute cost by using a trained DR many times. We can amortize compute costs due to the observed scale generalization: as our results show, a DR trained on 400M inner models generalizes to LLMs with parameters of 50M to 1B parameters.
>
>
> > Would Multi-criteria optimization be possible to also reduce dataset size directly?
>
> To target reducing the dataset size directly would indeed be possible; this would require introducing a different continuous relaxation of the main problem formulation (Eq 3), one which would operate at the dataset level instead of at the mini-batch level.
>
> Another way to use multi-criteria optimization is in the setting where one has a specific set of capabilities to target and corresponding dataset proxies. Then, the validation dataset on which the outer loss is calculated could include multiple such proxy datasets, for example, the validation dataset could include Wikipedia (as a proxy for high quality text), GitHub data (as a proxy for coding capability) or podcasts (as a proxy for conversational capability), etc. This would optimize a DR model for multiple criteria at the same time. The outer loss could be modified to explicitly make use of this (e.g. robustness against worst-case performance across the objectives).
>
>
> > What are the advantages over competing approaches mentioned in related works - e.g., influence functions, SHAP values, JST, ... ? More efficient? Better quality assessments?
>
> Influence functions (e.g. the TracIn algorithm from [1]) and SHAP values (e.g. as applied to deep learning in [2]) both use tabular representations (i.e. one score per datapoint). They also have myopic objectives, more closely related to a variant of our method with a single inner-loop unroll. In contrast, the DR is a scoring model that can generalise to new datapoints, and across model scales. Our meta-optimisation also accounts for multiple inner model steps, which may better characterise the value of the data.
>
> JST [3] performs a two stage filtering process, which relies on a held out set of low-quality data for its predictions. This is in contrast to our work, where the validation set can be sampled from the same distribution as the training set, as evidenced by all our results.
>
> Our paper contains a short comparison to these and related approaches (in Sec 4.1), which is due in part to the page limit constraints on the initial submission. In the final paper version we will expand this section to better contextualize our method.
>
> [1] Pruthi et al., Estimating Training Data Influence by Tracing Gradient Descent
>
> [2] Wang et al., Data Shapley in One Training Run
>
> [3] Zhang et al., Just Select Twice: Leveraging Low Quality Data to Improve Data Selection
>
>
> > T=2 inner models updates have been found to be best. Did the performance remain stable with more updates and steadily increase? Can you please give possible explanations for the good fit of T=2?
>
> Performance remains relatively stable when performing more inner updates, but static device memory (static HBM) consumption increases significantly, since memory linearly depends on the number of inner updates, even with MixFlow-MG, as it mainly addresses dynamic HBM. One could use more advanced techniques from MixFlow-MG (such as host offloading) to handle static memory however our experiments showed that T=2 or 4 achieved the best trade-off between the performance wins and cost of experiments. The good performance of shorter unrolls provides some evidence that the influence of a given training example on the overall optimisation trajectory is relatively localised.
>
>
> > How complex are the chosen datasets? Do the findings generalize to any complexity of text-based downstream task?
>
>
> The datasets we used (C4 and The Pile) are standard in the field and were chosen specifically for their varying degrees and types of complexity, which provides a robust testbed for DataRater.
> The Pile has a lot of compositional complexity, as it is composed of many distinct smaller corpora. These range from highly structured/technical sources (from arXiv) to more informal and conversational text (from YouTube).This diversity presents the kind of complex, mixed-quality scenario where an effective data curation is most valuable.
> C4 is derived from a broad crawl of the internet and contains the full spectrum of complexity of human knowledge as reflected in the internet. A key feature of C4 is the presence of significant noise and quality variation. Our results show that the DataRater is particularly effective on datasets with this type of complexity and variation.
> The downstream tasks (HellaSwag, SIQA, PIQA, ARC Easy, Commonsense QA) were chosen because they measure more than simple pattern recognition. These are challenging benchmarks that require sophisticated reasoning, and indicate whether a model has developed a nuanced understanding from its training data. We believe our findings generalise well to other text-based downstream tasks.
>
>
> > Is the optimization process sufficiently stable for updating a trained DataRater wrt new data?
>
>
> Yes. The DataRater utilizes a standard non-causal Transformer architecture. While optimization via meta-gradients involves a complex bilevel optimization landscape, we address the stability of this process explicitly. As described in Section 2, we employ several key techniques to stabilize meta-gradient computation, including: using a population of inner models (line 129); passing the meta-gradient for each inner model through an individual Adam optimizer (line 130) and averaging the resulting updates; and periodically reinitializing the inner models to stratify learning progress (lines 131-132).
>
> These stabilization mechanisms ensure robust optimization during initial training and are equally applicable when updating a trained DataRater on new data. Furthermore, updating the model (fine-tuning) begins from an already converged initialization, a process that is typically more stable than training from scratch. To further highlight this stability, we will include a plot of the meta-gradient update norms throughout training in the final submission.
>
>
> We hope we have addressed your concerns. Please let us know if you have any other questions.

---

### Comment · Area_Chair_T8L9 · 2025-08-05
**Reminder: Discussion and Final Justification**

Dear Reviewers,

As we approach the end of the author–reviewer discussion phase (Aug 6, 11:59pm AoE), I kindly remind you to read the author rebuttal carefully, especially any parts that address your specific comments.

Please consider whether the response resolves your concerns, and if not, feel free to engage in further discussion with the authors while the window is still open.

Your timely participation is important to ensure a fair and constructive review process. If you feel your concerns have been sufficiently addressed, you may also submit your Final Justification and update your rating early. Thank you for your contributions.

Best,

AC

---

### Note · Authors · 2025-08-15

We sincerely thank the reviewers for their insightful feedback and engaging discussions. We appreciate the reviewers found our work "very well written" (VwDw), "clear" (i2Zt, JMg5) while addressing a "critical and underexplored" (pnjF) challenge / an "extremely important problem" (VwDw). Reviewers also appreciated our qualitative results, finding them "cute" (pnjF) and suggesting that "more papers in the data curation literature should adopt this" (VwDw).

Our paper introduces the DataRater, a meta-learning approach to automated dataset curation. Our approach is found "sensible and original" (by i2Zt) and "technically sound" (by JMg5). A key novelty of our method is that it can be used in a self-supervised nature, learning the value of data directly -- without requiring a curated golden dataset, which is a significant advantage over existing methods.

Reviewers highlighted the significance of our empirical results, including the "substantial compute savings" (pnjF) achievable (of up to 46.6%) and the "consistent performance across model scales" (pnjF). Crucially, reviewers recognized our method's cross-scale generalization (pnjF), from 400M to 1B, which is "empirically robust" (JMg5) and enables amortizing the meta-training cost -- this was also denoted as "practically useful and rarely demonstrated in other works" (pnjF).

During the rebuttal, we have addressed all reviewers concerns, including discussions on meta-optimization stability, cost-benefit and complementary to existing filters. We will further strengthen our final paper by incorporating the discussed supplementary analyses. Specifically, in response to the panel's primary suggestion, we will add a perplexity-based filtering baseline to provide a more comprehensive comparison. To ensure the final paper is as clear and impactful as possible for readers, we will also incorporate the additional correlation analysis, and expand on the related work, meta-training stability and technical implementation details.

We believe the DataRater offers a scalable and principled alternative to manual data curation -- representing a meaningful step forward in automating current dataset curation pipelines.

---

### Decision · Program_Chairs · 2025-09-17

**Decision:**

Accept (poster)

**Comment:**

This work proposed the "DataRater", a meta-learning approach to automatically learn the value of data and use it to improve the compute efficiency of training foundation models. The DataRater framework trains a data-selector model in a meta-learning fashion by optimizing through unrolled inner optimizations of an actual LM being trained. The paper conducts experiments across different model scales on three publicly available datasets. Empirical evidence shows that the trained DataRater models can select high-quality training data for LMs, thereby improving compute efficiency of training.

This work has multiple significant strengths.

- The problem outlined by the paper---improving training efficiency via data curation---is an extremely important problem and and underexplored at scale.
- Meta-learning data value, bilevel optimization, and filtering via data scoring are all well-explored comprehensively.
- The paper is well-written with clear mathematical formulation and algorithmic specification.


While a reviewer still suggests that this work lacks empirical comparisons with heuristic, perplexity-based, or random filtering, so the key baseline gap and cost-benefit uncertainty remain. This does not change the consensus to accept this work.

The work touched an important problem and made significant progress with solid empirical results and clear theoretical motivation.

Given the consensus among the reviewers and the foregoing strengths, I recommend Accept (Poster).